# Effective Generation of Feasible Solutions for Integer Programming via Guided Diffusion

## Abstract

Feasible solutions are crucial for Integer Programming (IP) since they can substantially speed up the solving process. In many applications, similar IP instances often exhibit similar structures and shared solution distributions, which can be potentially modeled by deep learning methods. Unfortunately, existing deep-learning-based algorithms, such as Neural Diving (Nair et al., 2020), fail to capture the full underlying distributions and can only generate *partial* feasible solutions for a given IP instance. In this paper, we propose a novel framework that generates *complete* feasible solutions *end-to-end*. Our framework leverages contrastive learning to characterize the relationship between IP instances and solutions, and learns latent embeddings for both IP instances and their solutions. Further, the framework employs diffusion models to learn the distribution of solution embeddings conditioned on IP representations, with a dedicated guided sampling strategy that accounts for both constraints and objectives. We empirically evaluate our framework on four typical datasets of IP problems, and show that it effectively generates complete feasible solutions with a higher probability and a better quality for a given IP instance than the state-of-the-art.

## 1 Introduction

Integer Programming (IP) in the field of operation research is a class of optimization problems where some or all of the decision variables are constrained to be integers (Wolsey, 1998). Despite their importance in a wide range of applications such as production planning (Silver et al., 1998; Pochet & Wolsey, 2006), resource allocation (Katoh & Ibaraki, 1998), and scheduling (Toth & Vigo, 2002; Pantelides et al., 1995; Sawik, 2011), IP is known to be NP-hard and in general very difficult to solve. For decades, a significant effort has been made to develop sophisticated algorithms and efficient solvers, e.g., branch-and-bound (Lawler & Wood, 1966), cutting plane method (Kelley, 1960) and large neighborhood search algorithms (Pisinger & Ropke, 2019). These methods, however, can be computationally expensive because the search space for large-scale problems can be exponentially large. Moreover, these algorithms rely heavily on a feasible solution input that will crucially determine the whole search process. Hence, having a scalable method that produces feasible solutions for any IP instances is desirable for many real-world applications.

To generate feasible solutions, prior works (Nair et al., 2020; Han et al., 2023; Yoon, 2022) have advocated the employment of deep learning to capture similarity of the IP instances from the same domain in order to expedite solving. These works follow the method proposed by Gasse et al. (2019): model an IP instance using a bipartite graph to and then extracting variable representations of this graph with Graph Convolutional Networks (GCN). However, this approach has limitations in predicting solutions directly: it does not explicitly integrate objective and constraint information during the sampling process, leading to infeasible complete solutions. Nair et al. (2020); Yoon (2022); Han et al. (2023) thus focus on generating partial solutions by GCN, where only a subset of variables is assigned values using neural networks. Importantly, in many cases, the proportion of variables predicted by the neural network is set at a relatively low ratio (less than 50%) to ensure feasibility. Furthermore, such methods tend to be inefficient, primarily due to the introduction of auxiliary problems. For instance, the Completesol heuristic (Maher et al., 2017), a classical approach, solves an auxiliary integer programming model which is constructed by adding constraints to fix the variables from partial solutions. In another approach, Han et al. (2023) propose a predict-and-search framework which constructs a trust region to search for high-quality feasible solutions after acquir-

ing partial solutions. Nevertheless, this approach still necessitates a search within a neighborhood as an auxiliary problem. This prevailing landscape underscores the need for the development of an end-to-end deep learning framework to generate complete and feasible solutions for IP problems.

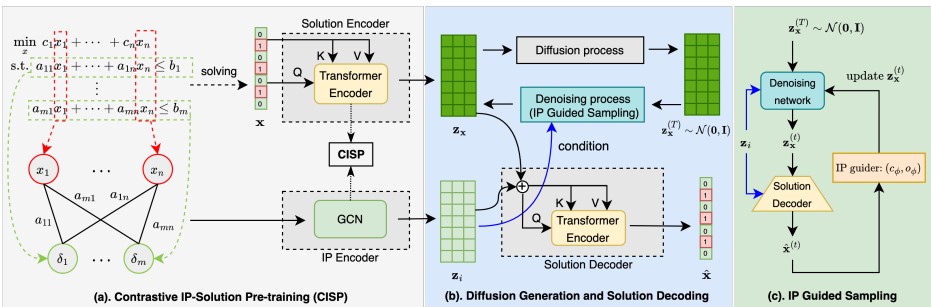

Figure 1: Our method first trains the IP Encoder and Solution Encoder to acquire the IP embeddings ($z_i$) and Solution embeddings ($z_x$) using CISP. We then jointly train diffusion models and the solution decoder to capture the distribution of solutions given a specific IP instance. In the sampling stage, we employ an IP guided diffusion sampling to account for both the objective and constraints.

Diffusion models (Ho et al., 2020; Sohl-Dickstein et al., 2015) have exhibited notable advantages in various generative tasks, primarily owing to their superior mode-coverage and diversity (Bayat, 2023). Notable applications include high-fidelity image generation (Dhariwal & Nichol, 2021), image-segmentation (Amit et al., 2021), and text-to-image synthesis (Ramesh et al., 2022). These successes motivates the launch of an investigation into harnessing the generative capability of diffusion models for acquiring feasible solutions of IP problems.

To this end, we introduce a comprehensive end-to-end generative framework presented in figure 1 to produce high-quality feasible solutions for IP problems. First of all, stemming inspiration from DALL.E-2 (Ramesh et al., 2022) for text-to-image translation, we employ a multimodal contrastive learning approach, akin to the CLIP Algorithm (Radford et al., 2021), to obtain embeddings for an IP instance $i$, denoted as $z_i$, and solution embeddings $z_x$ for solutions $x$ (Section 3). Subsequently, we employ DDPM (Ho et al., 2020) to model the distribution of $z_x$ conditioned on $z_i$ (Section 3). During this phrase, a decoder is concurrently trained with the task of solution reconstruction (Section 3). Finally, to enhance the quality of the feasible solutions during the sampling process, we propose the IP-guided sampling approaches tailored for both DDPM and DDIM (Song et al., 2021a) which explicitly consider both constraints and objectives during sampling. Our experimental results shown in Section 5 substantiate the efficacy of this approach in generating complete and feasible solutions for a given IP instance with a higher probability and better quality than the state-of-the-art.

## 2 BACKGROUND

**Integer Programming and Its Representations.** Integer programming (IP) is a class of NP-hard problems where the goal is to optimize a linear objective function, subject to linear and integer constraints. Without loss of generality, we focus on minimization which can be formulated as follows,

$$\min_{\mathbf{x}} \mathbf{c}^\top \mathbf{x} \qquad \text{subject to } \mathbf{A}\mathbf{x} \leq \mathbf{b}, \qquad \mathbf{x} \in \mathbb{Z}^n \qquad (1)$$

where $\mathbf{c} \in \mathbb{R}^n$ denotes the objective coefficient, $\mathbf{A} = [\mathbf{a}_1^\top, \mathbf{a}_2^\top, ..., \mathbf{a}_m^\top] \in \mathbb{R}^{m \times n}$ is the coefficient matrix of constraints and $\mathbf{b} = [b_1, b_2, ..., b_m]^\top \in \mathbb{R}^m$ represents the right-hand-side vector. For simplicity, we focus on binary integer variables, where $\mathbf{x}$ takes values in $\{0, 1\}^n$. This is a reasonable simplicity as any integer programming problem can be converted into a 0-1 programming problem (Dantzig, 1963). Throughout this paper, we adopt the term *IP instance* to denote a specific instance within the domain of some Integer Programming (IP) problem.

Bipartite graph representation, proposed by Gasse et al. (2019), is a commonly used and useful way to extract features of an IP instance for machine learning purposes. This representation, see the left part of Figure 1 (a) for an example, divides the constraints and variables into two different

sets of nodes, and uses a Graph Convolution Network (GCN) to learn the representation of nodes. Recently, Nair et al. (2020) proposed several changes to the architecture of GCN for performance improvements. Therefore, in this work, we use the bipartite graph structure combined with GCN to extract the embeddings of IP instances (see (Gasse et al., 2019; Nair et al., 2020) for more details).

**DDPM and DDIM.** Diffusion models learn a data distribution by reversing a gradual noising process. In the DDPM method (Ho et al., 2020), when presented with a data point sampled from an actual data distribution, denoted as $\mathbf{z}_{\mathbf{x}}^{(0)} \sim q(\mathbf{z}_{\mathbf{x}})$, a diffusion model, as described in Sohl-Dickstein et al. (2015); Ho et al. (2020), typically involves two distinct phases. In the forward process, a sequence of Gaussian noise is incrementally added to the initial sample over a span of $T$ steps, guided by a variance schedule denoted as $\beta_1, \beta_2, \dots, \beta_T$. This process yields a sequence of noisy samples $\mathbf{z}_{\mathbf{x}}^{(1)}, \mathbf{z}_{\mathbf{x}}^{(2)}, \dots, \mathbf{z}_{\mathbf{x}}^{(T)}$. Subsequently, the transition for the forward process can be described as: $q(\mathbf{z}_{\mathbf{x}}^{(t)}|\mathbf{z}_{\mathbf{x}}^{(t-1)}) = \mathcal{N}(\mathbf{z}_{\mathbf{x}}^{(t)}; \sqrt{1-\beta_t}\mathbf{z}_{\mathbf{x}}^{(t-1)}, \beta_t \mathbf{I})$. In fact, $\mathbf{z}_{\mathbf{x}}^{(t)}$ can be sampled at any time step $t$ in a closed form employing the notations $\alpha_t := 1 - \beta_t$ and $\bar{\alpha} := \prod_{s=1}^{t} \alpha_s$, $\mathbf{z}_{\mathbf{x}}^{(t)} = \sqrt{\bar{\alpha}_t}\mathbf{z}_{\mathbf{x}}^{(0)} + \sqrt{1-\bar{\alpha}_t}\boldsymbol{\epsilon}$, where $\boldsymbol{\epsilon} \sim \mathcal{N}(0, \mathbf{I})$. In the reverse process (denoising process), we need to model the distribution of $\mathbf{z}_{\mathbf{x}}^{(t-1)}$ given $\mathbf{z}_{\mathbf{x}}^{(t)}$ as a Gaussian distribution, which implies that $p_\theta(\mathbf{z}_{\mathbf{x}}^{(t-1)}|\mathbf{z}_{\mathbf{x}}^{(t)}) = \mathcal{N}\left(\mathbf{z}_{\mathbf{x}}^{(t-1)}; \boldsymbol{\mu}_\theta(\mathbf{z}_{\mathbf{x}}^{(t)}, t), \boldsymbol{\Sigma}_\theta(\mathbf{z}_{\mathbf{x}}^{(t)}, t)\right)$, where the variance $\boldsymbol{\Sigma}_\theta(\mathbf{z}_{\mathbf{x}}^{(t)}, t)$ can be fixed to a known constant Ho et al. (2020) or learned with a separate neural network (Nichol & Dhariwal, 2021), while the mean can be approximately computed by adding $\mathbf{z}_{\mathbf{x}}^{(0)}$ as a condition, $\boldsymbol{\mu}_\theta(\mathbf{z}_{\mathbf{x}}^{(t)}, t) = \frac{\sqrt{\alpha_t}(1-\bar{\alpha}_{t-1})}{1-\bar{\alpha}_t}\mathbf{z}_{\mathbf{x}}^{(t)} + \frac{\sqrt{\bar{\alpha}_{t-1}}\beta_t}{1-\bar{\alpha}_t}\mathbf{z}_{\mathbf{x}}^{(0)}$.

To improve the efficiency of sampling of DDPM, DDIM (Song et al., 2021a) formulates an alternative non-Markovian noising process with the same forward marginals as DDPM, but rewrites the probability $p_\theta(\mathbf{z}_{\mathbf{x}}^{(t-1)}|\mathbf{z}_{\mathbf{x}}^{(t)})$ in reverse process as a desired standard deviation $\sigma_t$. DDIM them derives the following distribution in the reverse process,

$$q_\sigma(\mathbf{z}_{\mathbf{x}}^{(t-1)}|\mathbf{z}_{\mathbf{x}}^{(t)}, \mathbf{z}_{\mathbf{x}}^{(0)}) = \mathcal{N}\left(\mathbf{z}_{\mathbf{x}}^{(t-1)}; \sqrt{\bar{\alpha}_t}\mathbf{z}_{\mathbf{x}}^{(0)} + \sqrt{1 - \bar{\alpha}_{t-1} - \sigma_t^2}\boldsymbol{\epsilon}^{(t)}, \sigma_t^2 \mathbf{I}\right), \quad (2)$$

where $\boldsymbol{\epsilon}^{(t)} = (\mathbf{z}_{\mathbf{x}}^{(t)} - \sqrt{\bar{\alpha}}\mathbf{z}_{\mathbf{x}}^{(0)})/(1 - \bar{\alpha})$ shows the direction pointing to $\mathbf{z}_{\mathbf{x}}^{(t)}$.

## 3 MODEL ARCHITECTURE

Our training dataset consists of pairs $(i, \mathbf{x})$ of IP instances $i$ and their corresponding feasible solutions $\mathbf{x}$. Given an instance $i$, let $\mathbf{z}_i \in \mathbb{R}^{n \times d}$ and $\mathbf{z}_{\mathbf{x}} \in \mathbb{R}^{n \times d}$ be the embeddings of the IP instance and the solution respectively, where $n$ is the number of variables and $d$ is the embedding dimension. It is worth noting that the same IP instance can have multiple different feasible solutions, meaning that we need to model the distribution of feasible solutions, which are discrete, by conditioning on a given IP instance. However, diffusion models, i.e. DDPMs (Ho et al., 2020; Sohl-Dickstein et al., 2015), mostly focus on continuous distribution. We thus use an encoder to transform the solutions $\mathbf{x}$ from a discrete space to a continuous embedding space $\mathbf{z}_{\mathbf{x}}$, and construct a diffusion model to learn the distribution of solutions given an IP embedding $\mathbf{z}_i$. Finally, a decoder is trained to recover the predicted solution $\hat{\mathbf{x}}$ from the embedding $\mathbf{z}_{\mathbf{x}}$. To effectively build the connection between the IP instance $i$ and solution $\mathbf{x}$, we first apply Contrastive IP-Solution Pre-training (CISP) module, similar to CLIP (Radford et al., 2021) which is used for text-to-image generation, to produce IP embedding $\mathbf{z}_i$ and solution embedding $\mathbf{z}_{\mathbf{x}}$. Overall, our model consists of three key components: a *Contrastive IP-Solution Pre-training (CISP) module* that produces IP embeddings $\mathbf{z}_i$ and solution embeddings $\mathbf{z}_{\mathbf{x}}$; a *diffusion module* $p(\mathbf{z}_{\mathbf{x}}|\mathbf{z}_i)$ that generates solution embedding $\mathbf{z}_{\mathbf{x}}$ conditioned on IP embedding $\mathbf{z}_i$; and a *decoder module* $p(\mathbf{x}|\mathbf{z}_{\mathbf{x}}, \mathbf{z}_i)$ that recovers solution $\mathbf{x}$ from embedding $\mathbf{z}_{\mathbf{x}}$ conditioned on IP embedding $\mathbf{z}_i$. We provide more details on each module in the following sections.

**Contrastive IP-Solution Pre-training.** Previous works (Nair et al., 2020; Han et al., 2023) show the crucial importance of establishing the connection between the IP instances and the solutions, and propose to implicitly learn this connection through the task of predicting feasible solutions. These approaches may not exhibit strong generalization capabilities on new instances because they only utilize the collected solutions in dataset without considering feasibility explicitly during training. To

more effectively capture this relationship, we propose to employ a contrastive learning task to learn representations for IP instances and embeddings for solutions by constructing feasible and infeasible solutions. The intuition behind is to ensure that the IP embeddings stay close to the embeddings of their feasible solutions, and away from the embeddings of the infeasible ones. To avoid explicitly constructing infeasible solutions, we proposed Contrastive IP-Solution Pre-training (CISP) algorithm, as opposed to Contrastive Language-Image Pre-training (CLIP) (Radford et al., 2021), to train IP encoder and solution encoder. Specifically, for IP encoder, we extract the representation of IP instances via a bipartite graph structure (Gasse et al., 2019), and use the structure of GCNs from Neural Diving (Nair et al., 2020) to generate all variables' embeddings as IP embeddings $\mathbf{z}_i$. For solution encoder, we use the encoder of the transformer to obtain the representations of each variable as solution embeddings $\mathbf{z}_\mathbf{x}$. Both $\mathbf{z}_\mathbf{x}$ and $\mathbf{z}_i$ have the same dimension to compute pairwise cosine similarities later. Since the number of variables in different IP instances may vary, we perform zero-padding on $\mathbf{z}_i$ and dummy-padding on $\mathbf{x}$ (i.e. padding 2 for 0-1 integer programming) to align the dimensions. The zero-padding for $\mathbf{z}_i$ is done to ensure that the cosine similarity remains unaffected. CISP algorithm then learns to maximize the similarity between embeddings of IP and corresponding solution pairs, and to minimize the similarity between the embeddings of incorrect pairs, which is achieved through optimizing a symmetric cross-entropy loss, as detailed in Appendix A.1.

**Diffusion Generation.** To leverage diffusion models for generating feasible solutions (discrete variables), we use the solution embedding $\mathbf{z}_\mathbf{x} \in \mathbb{R}^{n \times d}$ from the aforementioned CISP as the objective of generation. In addition, $\mathbf{z}_i$ is considered as a condition for generating high-quality results. According to Ho et al. (2020), we parameterize $p_\theta(\mathbf{z}_\mathbf{x}^{(t-1)}|\mathbf{z}_\mathbf{x}^{(t)}, \mathbf{z}_i) = \mathcal{N}\left(\mathbf{z}_\mathbf{x}^{(t-1)}; \boldsymbol{\mu}_\theta(\mathbf{z}_\mathbf{x}^{(t)}, \mathbf{z}_i, t), \boldsymbol{\Sigma}_\theta(\mathbf{z}_\mathbf{x}^{(t-1)}, \mathbf{z}_i, t)\right), \forall t \in [T, T-1, ..., 1]$ in reverse process, where $\mathbf{z}_\mathbf{x}^{(0)} = \mathbf{z}_\mathbf{x}$. Different from predicting the noise of each step in a general diffusion training phase, we predict $\mathbf{z}_\mathbf{x}$ directly as it empirically performs better. The training loss is defined as follows,

$$\mathcal{L}_{\text{MSE}} \triangleq \mathbb{E}_{t, \mathbf{z}_\mathbf{x}^{(t)}}\left[\|\mathbf{f}_\theta(\mathbf{z}_\mathbf{x}^{(t)}, \mathbf{z}_i, t) - \mathbf{z}_\mathbf{x}\|^2\right], \tag{3}$$

where $\mathbf{f}_\theta$ is an encoder-only transformer model and $\mathbf{z}_\mathbf{x}^{(t)} = \sqrt{\bar{\alpha}_t}\mathbf{z}_\mathbf{x} + \sqrt{1 - \bar{\alpha}_t}\boldsymbol{\epsilon}^{(t)}, \boldsymbol{\epsilon}^{(t)} \sim \mathcal{N}(\mathbf{0}, \mathbf{I})$. The specific model structure can be found in Appendix A.2.

**Solution Decoding.** The decoder $\mathbf{d}_\phi$ plays a crucial role in reconstructing the solution $\mathbf{x}$ from the solution embedding $\mathbf{z}_\mathbf{x}$. To enhance the robustness of the solution recovery, we jointly train the decoder $\mathbf{d}_\phi$ with the diffusion model. Specifically, we concatenate the solution embedding $\hat{\mathbf{z}}_\mathbf{x} = \mathbf{f}_\theta(\mathbf{z}_\mathbf{x}^{(t)}, \mathbf{z}_i, t)$ generated by the diffusion model with the IP embedding $\mathbf{z}_i$, and use the concatenated vector as input to a transformer encoder to obtain the reconstructed solution $\hat{\mathbf{x}} = \mathbf{d}_\phi(\hat{\mathbf{z}}_\mathbf{x}, \mathbf{z}_i)$. This process is associated with the cross-entropy loss defined as: $\mathcal{L}_{\text{CE}} \triangleq -\mathbb{E}_\mathbf{x}[\log \hat{\mathbf{x}}] = -\mathbb{E}_\mathbf{x}[\log \mathbf{d}_\phi(\hat{\mathbf{z}}_\mathbf{x}, \mathbf{z}_i)]$. To explicitly account for constraints in the training process, we introduce a penalty term to measure the degree of constraint violation. More specifically, let $\mathbf{a}_k^T$ be the $k$th row of matrix $\mathbf{A}$ in equation 1, the constraint violation (CV) loss is defined as $\mathcal{L}_{\text{CV}} \triangleq \frac{1}{m}\sum_{k=1}^m \max(\mathbf{a}_k^T\hat{\mathbf{x}} - b_k, 0)$, where $m$ is the number of constraints. The total loss for training diffusion and decoder therefore consists of the three parts:

$$\mathcal{L} = \mathcal{L}_{\text{MSE}} + \mathcal{L}_{\text{CE}} + \lambda\mathcal{L}_{\text{CV}}, \tag{4}$$

where $\lambda$ is a hyper-parameter to regulate the penalty. The full training procedure is given in Algorithm 2 in Appendix A.3 and the training details can be found in Appendix A.7.

## 4 IP GUIDED SAMPLING

Once the models have been trained, we can then sample variable assignments by running the sampling algorithm of DDPM or DDIM from a random Gaussian noise $\mathbf{z}_\mathbf{x}^{(T)} \sim \mathcal{N}(\mathbf{0}, \mathbf{I})$. Interestingly we find that, without suitable guidance, diffusion model is prone to generate inaccurate distributions, e.g. violating constraints for a given IP instance as shown in section 5.1. We thus consider the constraints information $(\mathbf{A}, \mathbf{b})$ and objective coefficient $\mathbf{c}$ during sampling. We present the IP guided diffusion sampling for both DDPM and DDIM, of which the latter is faster and better in terms of the quality and feasibility, as shown in Section 5.

### 4.1 IP GUIDED DIFFUSION SAMPLING

Consider a conditional diffusion model $p_\theta(\mathbf{z}_\mathbf{x}^{(t)}|\mathbf{z}_\mathbf{x}^{(t+1)}, \mathbf{z}_i)$, we first introduce *constraint guidance* by designing each transition probability as

$$p_{\theta,\phi}(\mathbf{z}_\mathbf{x}^{(t)}|\mathbf{z}_\mathbf{x}^{(t+1)}, \mathbf{z}_i, \mathbf{A}, \mathbf{b}) = Z p_\theta(\mathbf{z}_\mathbf{x}^{(t)}|\mathbf{z}_\mathbf{x}^{(t+1)}, \mathbf{z}_i)e^{-sc_\phi(\mathbf{z}_\mathbf{x}^{(t)}, \mathbf{z}_i, \mathbf{A}, \mathbf{b})}, \tag{5}$$

where $s$ is the gradient scale, $Z$ is a normalizing constant and $c_\phi(\mathbf{z}_\mathbf{x}^{(t)}, \mathbf{z}_i, \mathbf{A}, \mathbf{b}) = \sum_{k=1}^m \max(\mathbf{a}_k^T \mathbf{d}_\phi(\mathbf{z}_\mathbf{x}^{(t)}, \mathbf{z}_i) - b_k, 0)$ measures the violation of constraints. Let $\boldsymbol{\mu}$ and $\boldsymbol{\Sigma}$ be the mean and variance of the Gaussian distribution representing $p_{\theta,\phi}(\mathbf{z}_\mathbf{x}^{(t)}|\mathbf{z}_\mathbf{x}^{(t+1)}, \mathbf{z}_i)$. Then, $\log p_\theta(\mathbf{z}_\mathbf{x}^{(t)}|\mathbf{z}_\mathbf{x}^{(t+1)}, \mathbf{z}_i) = -\frac{1}{2}(\mathbf{z}_\mathbf{x}^{(t)} - \boldsymbol{\mu})^T \boldsymbol{\Sigma}^{-1}(\mathbf{z}_\mathbf{x}^{(t)} - \boldsymbol{\mu}) + C$, where $C$ is a constant. Consider the Taylor expansion for $c_\phi$ at $\mathbf{z}_\mathbf{x}^{(t)} = \boldsymbol{\mu}$,

$$c_\phi(\mathbf{z}_\mathbf{x}^{(t)}, \mathbf{z}_i, \mathbf{A}, \mathbf{b}) \approx c_\phi(\boldsymbol{\mu}, \mathbf{z}_i, \mathbf{A}, \mathbf{b}) + (\mathbf{z}_\mathbf{x}^{(t)} - \boldsymbol{\mu})\nabla_{\mathbf{z}_\mathbf{x}^{(t)}} c_\phi(\mathbf{z}_\mathbf{x}^{(t)}, \mathbf{z}_i, \mathbf{A}, \mathbf{b})\big|_{\mathbf{z}_\mathbf{x}^{(t)}=\boldsymbol{\mu}} = (\mathbf{z}_\mathbf{x}^{(t)} - \boldsymbol{\mu})\mathbf{g} + C_1,$$

where $\mathbf{g} = \nabla_{\mathbf{z}_\mathbf{x}^{(t)}} c_\phi(\mathbf{z}_\mathbf{x}^{(t)}, \mathbf{z}_i, \mathbf{A}, \mathbf{b})\big|_{\mathbf{z}_\mathbf{x}^{(t)}=\boldsymbol{\mu}}$ and $C_1$ is a constant. Similar to Classifier Guidance (Dhariwal & Nichol, 2021), we assume that $c_\phi(\mathbf{z}_\mathbf{x}^{(t)}, \mathbf{z}_i, \mathbf{A}, \mathbf{b})$ has low curvature compared to $\boldsymbol{\Sigma}^{-1}$ and thus have the following,

$$\log(p_{\theta,\phi}(\mathbf{z}_\mathbf{x}^{(t)}|\mathbf{z}_\mathbf{x}^{(t+1)}, \mathbf{z}_i, \mathbf{A}, \mathbf{b})) \approx -\frac{1}{2}(\mathbf{z}_\mathbf{x}^{(t)} - \boldsymbol{\mu})^T \boldsymbol{\Sigma}^{-1}(\mathbf{z}_\mathbf{x}^{(t)} - \boldsymbol{\mu}) - s(\mathbf{z}_\mathbf{x}^{(t)} - \boldsymbol{\mu})\mathbf{g} + sC_1 + C_2$$

$$= -\frac{1}{2}(\mathbf{z}_\mathbf{x}^{(t)} - \boldsymbol{\mu} + s\boldsymbol{\Sigma}\mathbf{g})^T \boldsymbol{\Sigma}^{-1}(\mathbf{z}_\mathbf{x}^{(t)} - \boldsymbol{\mu} + s\boldsymbol{\Sigma}\mathbf{g}) + C_3, \tag{6}$$

where $C_3$ is a constant and can be safely ignored. Therefore, $p_{\theta,\phi}(\mathbf{z}_\mathbf{x}^{(t)}|\mathbf{z}_\mathbf{x}^{(t+1)}, \mathbf{z}_i, \mathbf{A}, \mathbf{b})$ can be approximated by a Gaussian distribution with a mean shifted by $-s\boldsymbol{\Sigma}\mathbf{g}$. We can further inject the objective guidance to transition probability for acquiring high-quality solutions,

$$p_{\theta,\phi}(\mathbf{z}_\mathbf{x}^{(t)}|\mathbf{z}_\mathbf{x}^{(t+1)}, \mathbf{z}_i, \mathbf{A}, \mathbf{b}, \mathbf{c}) = Z p_\theta(\mathbf{z}_\mathbf{x}^{(t)}|\mathbf{z}_\mathbf{x}^{(t+1)}, \mathbf{z}_i)e^{-s\left((1-\gamma)c_\phi(\mathbf{z}_\mathbf{x}^{(t)}, \mathbf{z}_i, \mathbf{A}, \mathbf{b}) + \gamma o_\phi(\mathbf{z}_\mathbf{x}^{(t)}, \mathbf{z}_i, \mathbf{c})\right)}, \tag{7}$$

where $o_\phi(\mathbf{z}_\mathbf{x}^{(t)}, \mathbf{z}_i, \mathbf{c}) = \mathbf{c}^T \mathbf{d}_\phi(\mathbf{z}_\mathbf{x}^{(t)}, \mathbf{z}_i)$, $\mathbf{c}$ is the coefficient of objective from equation 1, and $\gamma \in [0, 1]$ is the leverage factor for balancing constraint and objective. The corresponding sampling method is called *IP Guided Diffusion Sampling*, as presented in Algorithm 3 in Appendix A.4.

### 4.2 NON-MARKOVIAN IP GUIDED SAMPLING

For the non-Markovian sampling scheme as used in DDIM, the method for conditional sampling is no longer invalid. To guide the sampling process, existing studies (Ho & Salimans, 2022; Dhariwal & Nichol, 2021) used the score-based conditioning trick from Song et al. (2021b) to construct a new epsilon prediction. We found that this trick turns out ineffective in our problem setting, possibly due to unique difficulties of finding feasible solutions in IP instances.

Instead, we find that adding a direction that guides the predicted solution to constraint region in each step of reverse process helps generate more reasonable solutions. Specifically, we first generate the predicted noise according to $\hat{\mathbf{z}}_\mathbf{x}^{(0)} = f_\theta(\mathbf{z}_\mathbf{x}^{(t)}, \mathbf{z}_i, t)$, $\boldsymbol{\epsilon}_\theta^{(t)} = \frac{\mathbf{z}_\mathbf{x}^{(t)} - \sqrt{\bar{\alpha}_t}\hat{\mathbf{z}}_\mathbf{x}^{(0)}}{\sqrt{1-\bar{\alpha}_t}}$. According to equation 2, the transition equation of DDIM for $\mathbf{z}_\mathbf{x}^{(t-1)}$ from a sample $\mathbf{z}_\mathbf{x}^{(t)}$ can be written as

$$\mathbf{z}_\mathbf{x}^{(t-1)} = \sqrt{\bar{\alpha}_t}\mathbf{f}_\theta(\mathbf{z}_\mathbf{x}^{(t)}, \mathbf{z}_i, t) + \sqrt{1 - \bar{\alpha}_{t-1} - \sigma_t^2}\boldsymbol{\epsilon}_\theta^{(t)} + \sigma_t\boldsymbol{\epsilon}_t. \tag{8}$$

where the first term is the prediction of $\mathbf{z}_\mathbf{x}^{(0)}$ and the second term is the direction pointing to $\mathbf{z}_\mathbf{x}^{(t)}$. To consider constraints in equation 1, we modify $\boldsymbol{\epsilon}_\theta^{(t)}$ by adding the direction of minimizing sum of constraint violation, that is

$$\hat{\boldsymbol{\epsilon}}_\theta^{(t)} = \boldsymbol{\epsilon}_\theta^{(t)} - s\nabla_{\mathbf{z}_\mathbf{x}^{(t)}} c_\phi(\mathbf{z}_\mathbf{x}^{(t)}, \mathbf{z}_i, \mathbf{A}, \mathbf{b}), \tag{9}$$

where $s$ is gradient scale. By replacing $\boldsymbol{\epsilon}_\theta^{(t)}$ in equation 8 with $\hat{\boldsymbol{\epsilon}}_\theta^{(t)}$, we obtain *Non-Makovian Constraint Guided Sampling*, which guides the solution generated in each transition to approach constraint region. Equivalently, it is to perform a gradient descent step with a step-size shrinking to zero

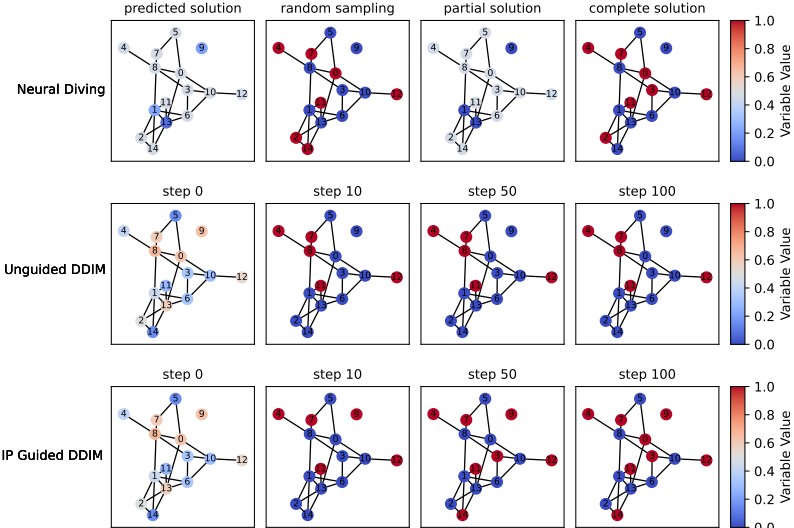

Figure 2: The sampling results from different methods. For Neural Diving, we present the predicted solution from GCN, random sampling according to the predicted solution, the partial solution obtained via SelectiveNet (only node 1, 9, 13 are assigned to 0), and the completing result by calling Completesol heuristic. For DDIM and IP Guided DDIM, we present the results from different time steps (transformed to solution space by a decoder) during sampling.

as $t \to 0$ when $\sigma_t \to 0$. To further consider the objective function together with the constraint, we can update $\epsilon_\theta^{(t)}$ as follows:

$$\hat{\epsilon}_\theta^{(t)} = \epsilon_\theta^{(t)} - s\nabla_{\mathbf{z}_\mathbf{x}^{(t)}}\bigg((1-\gamma)c_\phi(\mathbf{z}_\mathbf{x}^{(t)}, \mathbf{z}_i, \mathbf{A}, \mathbf{b}) + \gamma o_\phi(\mathbf{z}_\mathbf{x}^{(t)}, \mathbf{z}_i, \mathbf{c})\bigg), \tag{10}$$

where $\gamma \in [0, 1]$ is the leverage factor for balancing constraint and objective. This method is called *Non-Markovian IP Guided Diffusion Sampling*, as presented in Algorithm 4 in Appendix A.4.

## 5 EXPERIMENTS

This section empirically investigates the effectiveness of our method in solving IP instances. The efficacy is evaluated with two metrics: *feasible ratio* and *objective value*. The feasible ratio measures the proportion of feasible solutions among all sampled solutions and the objective value obtained from the generated feasible solutions measures the solution quality. We provide detailed description of the four datasets, i.e., Set Cover (SC), Combinatorial Auction (CA), Capacitated Facility Location (CF), and Independent Set (IS), along with baseline methods, Neural Diving (ND) (Nair et al., 2020), Predict-and-search algorithm (PS) (Han et al., 2023), SCIP 8.0.1 (Bestuzheva et al., 2023) and Gurobi 9.5.2 (Gurobi Optimization, 2021) in Appendix A.6. Additionally, we include the optimal objective values (obtained through running Gurobi for 100 seconds on each instance) as ground-truth. To ensure clarity, we use *IP Guided DDPM* to denote the (Markovian) IP Guided Diffusion sampling in Section 4.1, and *IP Guided DDIM* to represent the Non-Markovian IP Guided Diffusion sampling in Section 4.2. In the following, we first illustrate the guided diffusion sampling and emphasize its distinctions to Neural Diving and vanilla generation process in diffusion models in Section 5.1. Further, we evaluate the feasibility and quality of solutions generated by IP guided DDIM across all four datasets in Section 5.2. In Section 5.3, we demonstrate the scalability of our approach by applying it to larger instances and present the outcomes of qualitative analysis of solutions. Furthermore, we conduct an ablation study to investigate the impact of different guided approaches and contrastive learning in Appendix A.9. We use a workstation with two Intel(R) Xeon(R) Platinum 8163 CPU @ 2.50GHz, 176GB ram and two Nvidia V100 GPUs throughout the experiments. The detailed hyper-parameters for IP guided sampling can be found in Appendix A.10. We provide the total training and inference time in Appendix A.8.

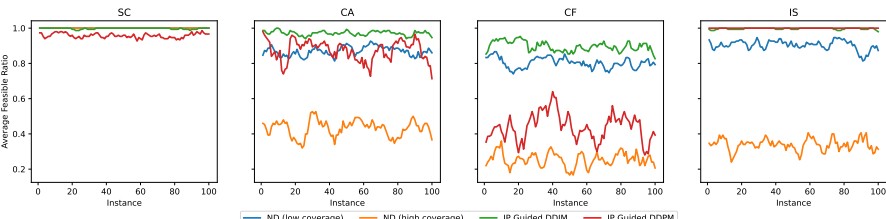

Figure 3: The feasible ratio in 100 instances, with each instance sampled 30 (partial) solutions.

| Instance | IP Guided DDIM | | IP Guided DDPM | | IP Guided DDIM + Completesol | | ND (low coverage) + Completesol | | PS + Gurobi | SCIP | Gurobi | Optimal |
|---|---|---|---|---|---|---|---|---|---|---|---|---|
| | obj. | fea. | obj. | fea. | obj. | fea. | obj. | fea. | obj. | obj. | obj. | obj. |
| SC (min) | 533.5 | 99.8% | 577.9 | 95.7% | **255.5** | **100%** | 849.0 | **100%** | 593.7 | 1967.0 | 522.4 | 168.28 |
| CA (max) | 26916.9 | 97.1% | 800.3 | 87.3% | **32491.1** | **99.7%** | 30143.6 | 87.0% | 31159.5 | 28007.4 | 30052.0 | 36102.6 |
| CF (min) | 25119.2 | 89.7% | 58488.1 | 44.0% | **14224.1** | **100%** | 14259.8 | 81.3% | 32119.8 | 84748.4 | 50397.3 | 11405.5 |
| IS (max) | 455.6 | 99.7% | 129.9 | **100%** | **639.4** | **100%** | 484.1 | 90.4% | 587.9 | 447.8 | 415.5 | 685.3 |

Table 1: The average objective value (obj.) and feasible ratio (fea.) for 100 instances.

## 5.1 ILLUSTRATIVE EXPERIMENTS

The maximal independent set problem involves finding the largest subset of nodes in an undirected graph where no two nodes are connected. We focus on an illustrative example: a graph consisting of 15 nodes and 22 edges. This graph can be transformed into an integer programming (IP) instance with 15 variables and 22 constraints. The graph is depicted in Figure 2, and we present the results from Neural Diving, and from the different time steps of unguided DDIM and IP Guided DDIM. Among the three algorithms, Neural Diving fixes only three node values and uses the Completesol heuristic to find an independent set containing seven nodes. However, the random sampling result from Neural Diving is infeasible. Unguided DDIM is unable to find a feasible solution. In contrast, IP Guided DDIM is able to fetch the optimal solution by finding an independent set containing eight nodes during the sampling process. Notably, the quality of the solution improves as the sampling process progresses. The independent set contains 5 nodes at step 10, 7 nodes at step 50, and finally 8 nodes (the optimal solution) at step 100. These indicate that IP Guided DDIM outperforms Neural Diving and Unguided DDIM in finding the optimal solution for this illustrative example.

## 5.2 PERFORMANCE EVALUATION

In this section, we evaluate the performance of different methods by comparing their average feasible ratios and average objective values across four datasets mentioned earlier. The average feasible ratios and objective values are computed for each method in 100 instances from the test set. For each instance, we sample 30 solutions and calculate the corresponding metrics, which allows us to assess the performance of each method in terms of both solution feasibility and objective value across the different datasets.

We first report the feasible ratio from different methods in all 4 datasets. We compare the feasibility ratio of the solutions generated by IP Guided Diffusion with those generated by ND. A prescribed coverage threshold $C$ is crucial in ND to ensure the feasibility of partial solutions, because a higher $C$ decreases the probability of generating feasible partial solutions in ND. Thus, we trained two variants of ND with different coverage thresholds in this experiment. For the SC, CA, and IS datasets, the coverage is set to 0.2 (low coverage) and 0.3 (high coverage) respectively. However, for the CF dataset, the coverage is set to 0.1 and 0.2 due to the difficulty in finding feasible partial solutions when $C > 0.2$. Figure 3 presents the average feasible ratio for 100 instances, where a higher ratio indicates better feasibility. In this comparison, IP guided DDIM outperforms other methods in terms of generating feasible solutions for all datasets.

To more comprehensively evaluate the performance of the diffusion model, we compare it against different baselines. In the case of ND, we use the low coverage model to prioritize the feasibility of partial solutions. We then apply the Completesol heuristic from SCIP to complete these partial

| Size | IP Guided DDIM | | IP Guided DDPM | | IP Guided DDIM + Completesol | | ND (low coverage) + Completesol | | PS + Gurobi | SCIP | Gurobi | Optimal |
|------|------|------|------|------|------|------|------|------|------|------|------|------|
| | obj. | fea. | obj. | fea. | obj. | fea. | obj. | fea. | obj. | obj. | obj. | obj. |
| small | 533.5 | 99.8% | 594.7 | 96.5% | **255.5** | **100%** | 849.0 | **100%** | 593.7 | 1967.0 | 522.4 | 168.3 |
| medium | 486.8 | 99.9% | 451.8 | 83.7% | **217.4** | **100%** | 1145.8 | **100%** | 737.0 | 2236.2 | 718.8 | 140.4 |
| large | 464.9 | **100%** | 440.7 | 77.9% | **195.9** | **100%** | 1465.6 | **100%** | 994.9 | 2386.0 | 1454.5 | 126.9 |

Table 2: The average objective value (obj.) and the feasible ratio (fea.) for 100 instances in 3 different size SC datasets.

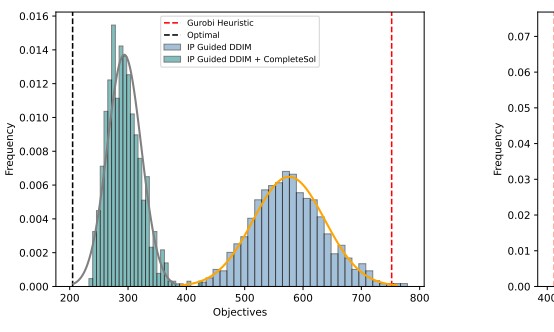
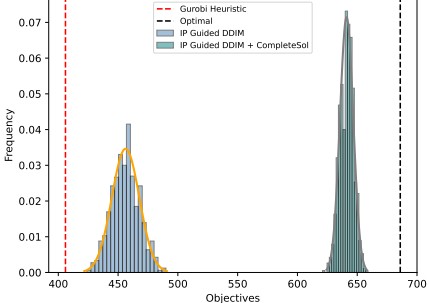

(a) SC instance (minimization).  (b) IS instance (maximization).

Figure 4: The objective distribution of 1000 solutions sampled from a single instance.

solutions and measure their objective values. In the case of the PS algorithm, we construct a trust region using partial solutions with the same proportion of assigned variables as ND and use Gurobi as the Solver for its modified instances, with the best heuristic solutions found as a benchmark. For SCIP, we adopt the first solution obtained through non-trivial heuristic algorithms during the solving phase. For Gurobi, we use the best heuristic solution for each instance as a benchmark. These approaches guarantee reasonable initial solutions to compare against. Our methods are able to generate complete feasible solutions, and can produce partial solutions by randomly sampling a certain proportion of variables from the complete solutions. For a fair comparison, we report the quality of complete solutions generated by our methods (IP Guided DDIM/DDPM) and the quality of partial solutions sampled from the complete solutions. We then use CompleteSol to fill in the remaining variables (IP Guided DDIM + CompleteSol). The partial solutions have the same expected proportion of assigned variables (coverage) as ND. The results are presented in Table 1. Clearly, the combination of partial solutions from IP guided DDIM and the Complete heuristic outperforms all other methods in terms of both feasible ratio and objective values. Furthermore, solutions generated through IP guided diffusion sampling show a 10% improvement in feasible ratios for the CA, CF, and IS datasets. For the SC dataset, all methods exhibit comparable feasible ratios. In contrast, in the CF and IS datasets, IP guided diffusion also generates solutions of better quality than both the heuristic solutions of SCIP and Gurobi.

## 5.3 SCALABILITY TEST AND QUALITATIVE ANALYSIS

Practitioners often aim to apply the models learned to solve problems of larger scales than the ones used for data collection and training. To estimate how well a model can generalize to bigger instances, we evaluate its performance on datasets of varying sizes from the Set Cover problem (minimization problem). We utilize three size categories: small (2000 variables, 1000 constraints), medium (3000 variables, 1500 constraints), and large (4000 variables, 2000 constraints). It is worth noting that all models were trained using the small-size dataset. The results in table 2 demonstrate that IP guided DDPM consistently performs well across all three different-sized datasets, indicating that our framework possesses strong generalization capabilities.

Diffusion models are generative models that capture the distribution of a dataset. In this experiment, we focus on the distribution of solutions generated by our methods. We take a single instance from

the SC dataset and IS dataset and use the IP Guided DDIM algorithm to generate 1000 complete feasible solutions. We also randomly sample 20% of variable values from each solution and use the CompleteSol heuristic to fill in the remaining variables. We analyze the distribution of objective values for these complete solutions and compare them with the optimal objective values and the best heuristic solutions found by the Gurobi heuristic. The results are shown in Figure 4. Clearly, the complete solutions directly generated by the IP Guided DDIM algorithm are superior to the best solutions from the Gurobi heuristic. Furthermore, the objective values from the partial solutions completed by the CompleteSol heuristic are closer to the optimal value.

## 6  RELATED WORK

We start the related work section with a summary of deep learning techniques used in the construction of feasible solutions for Integer Programming (IP) problems. Gasse et al. (2019) propose a method that combines a bipartite graph with a Graph Convolutional Network (GCN) to extract representations of Integer Programming (IP) instances. Although this approach is primarily employed to learn the branching policy in the branch and bound algorithm, it is worth noting that this modeling method can also be utilized for the prediction of solutions for IP instances. However, the solutions produced by GCN directly are often infeasible or sub-optimal. To address this, Neural Diving (Nair et al., 2020) leverages the SelectiveNet (Geifman & El-Yaniv, 2019) to assign values to only a subset of the variables based on a coverage threshold, with the rest of the variables being completed via an IP Solver. To further improve the feasibility of generated solutions, Han et al. (2023) proposes a predict-search framework that combines the predictions from GNN model with trust region method. However, this method still relies on IP solver to solve a modified instance (adding neighborhood constraints to origin instance) in order to get complete solutions. Similar to Nair et al. (2020), Khalil et al. (2022) integrate GNN into integer programming solvers and apply it to construct a partial solutions through a prescribed rounding threshold, which is then completed using SCIP. In contrast, our method aims to learn the latent structure of IP instances by diffusion models, and obtains complete feasible solutions through guided diffusion sampling, without any reliance on the IP solver. In Appendix A.12, we discuss other relevant studies in the field of neural networks for solving integer programming problems.

Another set of related works to our paper is diffusion models (Sohn et al., 2015; Ho et al., 2020). As the latest state-of-the-art family of deep generative models, diffusion models have demonstrated their ability to enhance performance across various generative tasks (Yang et al., 2022). In this paper, we focus our discussion specifically on conditional diffusion models. Unlike unconditional generation, conditional generation emphasizes application-level contents as a condition to control the generated results based on predefined intentions. To enable this conditioning, Dhariwal & Nichol (2021) introduce the concept of classifier guidance, which enhances sample quality by conditioning the generative process on an additional trained classifier. In the same vein, Ho & Salimans (2022) propose a joint training strategy for both conditional and unconditional diffusion models, i.e., classifier-free guidance. This approach combines the resulting conditional and unconditional scores to achieve a balance between sample quality and diversity. This idea has also found its effectiveness in Topology Optimization (Mazé & Ahmed, 2023). We thus take a similar derivation from the classifier guidance and devise IP-guided diffusion sampling by incorporating the objectives and constraints into the transition probability.

## 7  CONCLUSION

In this paper, we presented a comprehensive framework for generating feasible solutions for Integer Programming (IP) problems. We utilized the CISP approach to establish a link between IP instances and their corresponding solutions, allowing to obtain IP embeddings and solution embeddings. To effectively capture the distribution of feasible solutions, we leveraged diffusion models, which are known for their powerful learning capabilities, to learn the distribution of solution embeddings. We further employed a solution decoder to reconstruct the solutions from their embeddings. Importantly, we proposed an IP guided sampling algorithm that explicitly incorporates the objective and constraint information to generate high-quality solutions. The experimental results on four distinct datasets demonstrate the superiority of our approach to the state-of-the-art.

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

# A APPENDIX

## A.1 CISP ALGORITHM

The study conducted by Radford et al. (2021) underscores the substantial efficacy of contrasting pre-training in capturing multi-modal data, with particular emphasis on its application in the text-to-image transfer domain. Drawing inspiration from this seminal work, we introduce the CISP algorithm. The primary objective of CISP is to facilitate the learning of the IP Encoder $\mathbf{E}_I$ and the Solution Encoder $\mathbf{E_X}$, as illustrated in Algorithm 1. Within the scope of our investigation, we work with a mini-batch of data comprising instances denoted as $I$ and their corresponding solutions denoted as $\mathbf{X}$. The batch's bipartite graph representing instances $I$ is denoted as $\mathbf{G}$. We use $\mathbf{z}_I$ and $\mathbf{z_X}$ to denote the embeddings of instances and solutions, respectively. Notably, both $\mathbf{z}_I$ and $\mathbf{z_X}$ possess identical dimensions, enabling us to compute their cosine similarity. Furthermore, within the mini-batch, we use $\mathbf{z}_{I,j}$ and $\mathbf{z}_{\mathbf{X},k}$ to reference the $j$th sample in $\mathbf{z}_I$ and the $k$th sample in $\mathbf{z_X}$, respectively. Within this conceptual framework, we leverage the matrix $\mathbf{s} \in \mathbb{R}^{N \times N}$ to represent the similarity between $N$ instances and $N$ solutions. Each element $\mathbf{s}_{j,k}$, where $j, k \in \{1, ..., N\}$, corresponds to the logit employed in computation of the symmetric cross-entropy loss.

---

**Algorithm 1** Contrastive IP-Solutioin Pre-Training (CISP)

---

**Input**: The mini-batch size $N$, the mini-batch bipartite graph representations of IP instance set $I$, denoted by $\mathbf{G}$, and corresponding mini-batch solutions $\mathbf{X}$
**Require**: IP Encoder $\mathbf{E}_I$, Solution Encoder $\mathbf{E_X}$, temperature parameter $\tau$

  1: Get IP and solution embeddings $\mathbf{z}_I, \mathbf{z_X} = \mathbf{E}_I(\mathbf{G}), \mathbf{E_X}(\mathbf{X})$ *// $N \times n \times d$, where n is the padding length of variables and d is the embedding size.*
  2: **for** $j \in \{1, 2, ..., N\}$ and $k \in \{1, 2, ..., N\}$ **do**
  3:    Flatten $\mathbf{z}_{I,j}$ and $\mathbf{z}_{\mathbf{X},k}$ into vectors $\bar{\mathbf{z}}_{I,j}$ and $\bar{\mathbf{z}}_{\mathbf{X},k}$
  4:    $\mathbf{s}_{j,k} = e^\tau \cdot \bar{\mathbf{z}}_{I,j}^T \bar{\mathbf{z}}_{\mathbf{X},k}/(\|\bar{\mathbf{z}}_{I,j}\|\|\bar{\mathbf{z}}_{\mathbf{X},k}\|)$ *// compute similarity for IP and solution embeddings*
  5: **end for**
  6: Set labels $\mathbf{y} = (1, 2, ..., N)$
  7: Compute cross-entropy loss $\mathcal{L}_I$ by utilizing $\mathbf{s}_{j,*}$ and $\mathbf{y}$.
  8: Compute cross-entropy loss $\mathcal{L_X}$ by utilizing $\mathbf{s}_{*,k}$ and $\mathbf{y}$.
  9: Compute the symmetric loss $\mathcal{L} = (\mathcal{L}_I + \mathcal{L_X})/2$
10: **return** $\mathcal{L}$

---

## A.2 DIFFUSION MODEL STRUCTURE

The diffusion model is a transformer encoder-based model, as illustrated in Figure 5. At step $t$, we concatenate the IP embedding $\mathbf{z}_i$ with the noised solution embedding $\mathbf{z}_{\mathbf{x}}^{(t)}$ and sinusoidal timestep embeddings generated from timestep $t$. Subsequently, we utilize the Transformer Encoder to obtain the predicted solution $\hat{\mathbf{z}}_{\mathbf{x}}$. Moreover, we use masks in transformers to skip the computation of the padded variable components.

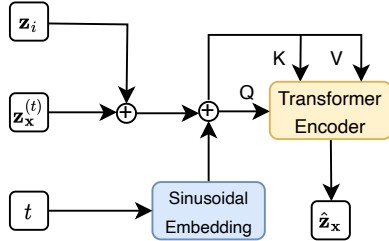

Figure 5: Diffusion model $\mathbf{f}_\theta(\mathbf{z}_{\mathbf{x}}^{(t)}, \mathbf{z}_i, t)$

## A.3 Training Diffusion and Solution Decoder

In this section, we present the procedure to train diffusion and decoder models simultaneously.

---

**Algorithm 2** Training diffusion and solution decoder

---

**Require**: IP instance embedding $\mathbf{z}_i$ from CISP, solution embedding $\mathbf{z}_\mathbf{x}$ from CISP, diffusion model $\mathbf{f}_\theta$, and solution decoder $\mathbf{d}_\phi$.

1: **repeat**
2:    $t \sim \text{Uniform}(\{1, ..., T\})$
3:    $\boldsymbol{\epsilon} \sim \mathcal{N}(\mathbf{0}, \mathbf{I})$
4:    $\hat{\mathbf{z}}_\mathbf{x} \leftarrow \mathbf{f}_\theta \left( \sqrt{\bar{\alpha}_t}\mathbf{z}_\mathbf{x} + \sqrt{1 - \bar{\alpha}_t}\boldsymbol{\epsilon}, \mathbf{z}_i, t \right)$ *// predicted $\hat{\mathbf{z}}_\mathbf{x}$*
5:    $\hat{\mathbf{x}} \leftarrow \mathbf{d}_\phi(\hat{\mathbf{z}}_\mathbf{x}, \mathbf{z}_i)$ *// reconstructed solution $\hat{\mathbf{x}}$*
6:    Take gradient descent step to minimize total loss in equation 4
7: **until** Reaches a fixed number epochs or satisfies an early stopping criteria

---

## A.4 IP Guided Sampling Algorithms

The IP guided diffusion sampling algorithms for both DDPM and DDIM are presented in Algorithm 3 and Algorithm 4, respectively.

---

**Algorithm 3** IP Guided Diffusion Sampling

---

**Input**: gradient scale $s$, leverage factor $\gamma$, constraint information $(\mathbf{A}, \mathbf{b})$ and objective coefficient $\mathbf{c}$
**Require**: diffusion model $\mathbf{f}_\theta$, solution decoder $\mathbf{d}_\theta$

1: $\mathbf{z}_\mathbf{x}^{(T)} \sim \mathcal{N}(\mathbf{0}, \mathbf{I})$
2: **for** $t$ from $T$ to $1$ **do**
3:    $\boldsymbol{\mu} \leftarrow \frac{\sqrt{\alpha_t}(1 - \bar{\alpha}_{t-1})}{1 - \bar{\alpha}_t}\mathbf{z}_\mathbf{x}^{(t)} + \frac{\sqrt{\bar{\alpha}_{t-1}}\beta_t}{1 - \bar{\alpha}_t}\mathbf{f}_\theta(\mathbf{z}_\mathbf{x}^{(t)}, \mathbf{z}_i, t)$
4:    $\boldsymbol{\Sigma} \leftarrow \boldsymbol{\Sigma}_\theta(\mathbf{z}_\mathbf{x}^{(t-1)}|\mathbf{z}_\mathbf{x}^{(t)}, \mathbf{z}_i)$
5:    $\boldsymbol{\mu} \leftarrow \boldsymbol{\mu} - s\boldsymbol{\Sigma}\nabla_{\mathbf{z}_\mathbf{x}^{(t-1)}} \left( (1 - \gamma)c_\phi(\mathbf{z}_\mathbf{x}^{(t)}, \mathbf{z}_i, \mathbf{A}, \mathbf{b}) + \gamma o_\phi(\mathbf{z}_\mathbf{x}^{(t)}, \mathbf{z}_i, \mathbf{c}) \right) \Big|_{\mathbf{z}_\mathbf{x}^{(t)} = \boldsymbol{\mu}}$
6:    $\mathbf{z}_\mathbf{x}^{(t)} \sim \mathcal{N}(\boldsymbol{\mu}, \boldsymbol{\Sigma})$
7: **end for**
8: **return** $\mathbf{d}_\phi(\mathbf{z}_\mathbf{x}^{(0)}, \mathbf{z}_i)$.

---

**Algorithm 4** Non-Markovian IP Guided Diffusion Sampling

---

**Input**: gradient scale $s$, leverage factor $\gamma$, constraint information $(\mathbf{A}, \mathbf{b})$ and objective coefficient $\mathbf{c}$
**Require**: diffusion model $\mathbf{f}_\theta$, solution decoder $\mathbf{d}_\theta$

1: $\mathbf{z}_\mathbf{x}^{(T)} \sim \mathcal{N}(\mathbf{0}, \mathbf{I})$
2: **for** $t$ from $T$ to $1$ **do**
3:    $\boldsymbol{\epsilon}_\theta^{(t)} \leftarrow (\mathbf{z}_\mathbf{x}^{(t)} - \sqrt{\bar{\alpha}_t}\mathbf{f}_\theta(\mathbf{z}_\mathbf{x}^{(t)}, \mathbf{z}_i, t))/\sqrt{1 - \bar{\alpha}_t}$.
4:    $\hat{\boldsymbol{\epsilon}}_\theta^{(t)} \leftarrow \boldsymbol{\epsilon}_\theta^{(t)} - s\nabla_{\mathbf{z}_\mathbf{x}^{(t)}} \left( (1 - \gamma)c_\phi(\mathbf{z}_\mathbf{x}^{(t)}, \mathbf{z}_i, \mathbf{A}, \mathbf{b}) + \gamma o_\phi(\mathbf{z}_\mathbf{x}^{(t)}, \mathbf{z}_i, \mathbf{c}) \right)$
5:    $\mathbf{z}_\mathbf{x}^{(t-1)} \leftarrow \sqrt{\bar{\alpha}_t}\mathbf{f}_\theta(\mathbf{z}_\mathbf{x}^{(t)}, \mathbf{z}_i, t) + \sqrt{1 - \bar{\alpha}_{t-1} - \sigma_t^2}\hat{\boldsymbol{\epsilon}}_\theta^{(t)} + \sigma_t\boldsymbol{\epsilon}_t$
6: **end for**
7: **return** $\mathbf{d}_\phi(\mathbf{z}_\mathbf{x}^{(0)}, \mathbf{z}_i)$.

---

## A.5 Feature Descriptions For Variables Nodes, Constraint Nodes And Edges

In Table 3, we provide a description of the features that are extracted using the *Ecole* library (Prouvost et al., 2020) and used as IP bipartite graph representations for training the GCN model.

| | Feature | Description |
|---|---|---|
| Variable | type | Type(binary, integer, impl. integer, continuous) as a one-hot encoding. |
| | coef | Objective coefficient, normalized. |
| | has_lb | Lower bound indicator. |
| | has_ub | Upper bound indicator. |
| | sol_is_at_lb | Solution value equals lower bound. |
| | sol_is_at_ub | Solution value equals upper bound. |
| | sol_frac | Solution value fractionality. |
| | basis_status | Simplex basis status(lower, basic, upper, zero) as a one-hot encoding. |
| | reduced_cost | Reduced cost, normalized. |
| | age | LP age, normalized. |
| | sol_val | Solution value. |
| Constraint | obj_cos_sim | Cosine similarity with objective. |
| | bias | Bias value, normalized with constraint coefficients. |
| | is_tight | Tightness indicator in LP solution. |
| | dualsol_val | Dual solution value, normalized. |
| | age | LP age, normalized with total number of LPs. |
| Edge | coef | Constraint coefficient, normalized per constraint. |

Table 3: Description of the variable, constraint and edge features in our bipartite graph representations.

## A.6 DATASETS AND BASELINES

### A.6.1 DATASETS.

We use four self-generated IP datasets from the Ecole library (Prouvost et al., 2020):

- *Set Cover* (SC) is to find the least number of subsets that cover a given universal set.
- *Combinatorial Auction* (CA) is to help bidders place unrestricted bids for bundles of goods and the aim is to maximize the revenue.
- *Capacitated Facility Location* (CF) is to locate a number of facilities to serve the sites with a given demand and the aim is minimize the total cost.
- *Independent Set* (IS) is to find the maximum subset of nodes of an undirected graph such that no pair of nodes are connected.

For all datasets, we randomly generate 1000 instances (800 for training, 100 for validation and 100 for testing). Table 4 summarizes the numbers of constraints, variables and problem type of each dataset. We then collect feasible solutions and their objective values for each instance by running the Gurobi (Gurobi Optimization, 2021) or SCIP Bestuzheva et al. (2023), where the time limit is set to 1000s for each instance. For those instances with a large number of feasible solutions, we only keep 500 best solutions. We adopt the same features exacted via the Ecole library as in (Gasse et al., 2019) and exclude those related to feasible solutions. The specific features are shown in Appendix A.5.

### A.6.2 BASELINES.

We compared our method with the following baselines:

- *SCIP* (Bestuzheva et al., 2023) (an open source solver): SCIP is currently one of the fastest non-commercial solvers for mixed integer programming (MIP) and mixed integer nonlinear programming (MINLP). Here, we focus on comparing the quality of initial solutions with SCIP. Instead of relying on the first feasible solution generated by SCIP, which are often of low quality due to the use of trivial heuristics, we employ the first solution produced via non-trivial heuristic algorithms during the solving process of SCIP (Bestuzheva et al., 2023) (i.e. the first feasible solution after the pre-solving stage of SCIP).
- *Gurobi* (Gurobi Optimization, 2021) (the powerful commercial solver): Gurobi is a highly efficient commercial mathematical optimization solver. As our focus is on comparing fea-

| Dataset | Constraints | Variables | Problem Type |
|---------|-------------|-----------|--------------|
| SC | 1000 | 2000 | minimize |
| CA | 786 | 1500 | maximize |
| CF | 5051 | 5050 | minimize |
| IS | 6396 | 1500 | maximize |

Table 4: Instance size of each dataset

sible solutions, we consider the best solutions obtained through Gurobi's default heuristic algorithms as a benchmark.

- *Neural Diving (ND)* (Nair et al., 2020): Neural Diving adopts a solution prediction approach, training a GCN to predict the value of each variable. It then incorporates SelectiveNet (Geifman & El-Yaniv, 2019) to generate partial solutions with a predefined coverage threshold $C$ (e.g., a coverage threshold $C = 0.2$ means that the expectation of the number of assigned variables by the neural network is 20%). This threshold is typically set to be low ($<0.5$) to ensure the feasibility of partial solutions. In our experiments, to evaluate the feasibility of solutions, we set two different coverage levels, namely low coverage (which usually indicates higher feasibility) and high coverage (which usually indicates lower feasibility), for each dataset. We then compare the feasibility of the partial solutions with the complete solutions generated by our methods. To assess the quality of solutions, we employ the Completesol heuristic in SCIP (Algorithm 4 in Maher et al. (2017)) to enhance the partial solutions. This heuristic involves solving auxiliary IP instances by fixing the variables from the partial solutions. By utilizing this heuristic, we can obtain more complete solutions and evaluate their quality.

- *Predict-and-search algorithm (PS)* (Han et al., 2023): The PS algorithm, similar to Neural Diving, utilizes graph neural networks to predict the value of each variable. It then searches for the best feasible solution within a trust region constructed by the predicted partial solutions. This method requires setting parameters $(k_0, k_1)$ to represent the numbers of 0's and 1's in a partial solution, and $\Delta$ to define the range of the neighborhood region of the partial solution. To search a high-quality feasible solution, this method adds neighborhood constraints to origin instance, which produces modified IP instance. Therefore, an IP solver such as SCIP or Gurobi is required to solve the modified instance and obtain feasible solutions. In our experiments, we use Gurobi as the solver and control the parameters $\Delta$ to ensure that the modified instance is 100% feasible. We considered the best heuristic solutions from the modified instance found by Gurobi as our baseline.

## A.7 TRAINING DETAILS

We trained the CISP and diffusion model on four IP datasets. Each dataset contained 800 training instances, with 500 solutions collected for each instance. In each batch, we sampled $64$ instances, and for each instance, We sample one solution from 500 solutions in proportion to the probability of the objective value. This implies that solutions with better objective values had a higher probability of being sampled. We iterated through all instances (with one solution per instance) in each epoch.

For the Solution Encoder, we utilized a single transformer encode layer with a width of 128. The IP encoder adopted the architecture described in Nair et al. (2020), using GCN to obtain embeddings for all variables as IP embeddings. Both models transformed the features of the solution and IP into latent variables with a dimension of 128, enabling convenient computation of cosine similarity in CISP. The CISP was trained using the AdamW Optimizer (Loshchilov & Hutter, 2019). We employed a decreasing learning rate strategy, starting with a learning rate of 0.001 and linearly decaying it by a factor of 0.9 every 100 epochs until reaching 800 epochs. The model training was performed with a batch size of 64.

For the diffusion model, we utilized a single-layer Transformer encoder with a width of 128 to predict $\mathbf{z_x}$ and adjusted the number of time steps to 1000. The forward process variances were set as constants, increasing linearly from $\beta_1 = 10^{-4}$ to $\beta_T = 0.02$, following the default setting of DDPM (Ho et al., 2020). The solution decoder model was jointly trained with the diffusion model

and consisted of two Transformer encode layers with a width of 128. The loss function was defined as the sum of the diffusion loss, decoder loss, and the penalty for violating constraints, as shown in equation 4. Here, $\lambda$ is set to be the number of variables in the instances from the training set, excluding the IS dataset, where $\lambda = 0$. We trained diffusion and decoder model for 100 epochs with batch size of 32 via Adam Optimizer (Kingma & Ba, 2015).

## A.8 TRAINING AND INFERENCE TIME

In this section, we report the training time (including CISP pretraining and Diffusion model) for each dataset, which takes 100 epochs for both CISP and Diffusion model to converge. Additionally, we provide the total inference time for sampling 3000 solutions by using IP Guided DDIM and DDPM. From Table 5, we observe that our method requires a reasonable amount of time for model training. During the inference phase, IP Guided DDIM demonstrates faster performance compared to IP Guided DDPM with average time of 0.46s-1.68s for sampling each solution. Moreover, as shown in the experiment results from Section 5, IP Guided DDIM also achieves better performance than DDPM, making it suitable for practical applications.

| Dataset | Training (CISP + Diffusion) | IP Guided DDIM | IP Guided DDPM |
|---------|-----------------------------|----------------|----------------|
| SC | 24.4m | 37.5m | 374m |
| CA | 9.3m | 23m | 233.5m |
| CF | 71.7m | 84m | 805m |
| IS | 11.1m | 23m | 234m |

Table 5: Total training time and inference time for sampling 3000 solutions for each dataset

## A.9 ABLATION STUDY

We ablate on unguided DDIM (with $s = 0$), constraint guided DDIM (with $\gamma = 1$), objective guided DDIM (with $\gamma = 0$) models and IP guided DDIM on four datasets. We also include an experiment where we train IP and solution embeddings directly via algorithm 2 without CISP, in order to assess the advantages of contrastive learning, i.e. IP Guided DDIP w/o CISP in Table 6. The results are presented in Table 6. Evidently, the constraint guidance is crucial in generating feasible solutions, and the objective guidance further enhances the quality of solutions. Moreover, the experiments demonstrate that CISP plays a crucial role in ensuring that the solutions produced by our methods are more feasible. Therefore, combining both constraint and objective guidance achieves good quality solutions with high probability.

| dataset | Unguided DDIM | | Constraint Guided DDIM | | Objective Guided DDIM | | IP Guided DDIM w/o CISP | | IP Guided DDIM | |
|---------|------|------|---------|--------|------|------|---------|-------|---------|--------|
| | obj. | fea. | obj. | fea. | obj. | fea. | obj. | fea. | obj. | fea. |
| SC (min) | - | 0.0% | 63046.9 | **99.8%** | - | 0.0% | 763.4 | 99.8% | **533.5** | **99.8%** |
| CA (max) | - | 0.0% | 5157.2 | **99.7%** | - | 0.0% | 23383.3 | 57.7% | 26916.9 | 97.1 % |
| CF (min) | - | 0.0% | 53311.2 | 74.1 % | - | 0.0% | 31319.8 | 41.7% | **25119.2** | **89.7%** |
| IS (max) | - | 0.0% | 386.5 | **100 %** | - | 0.0% | **479.1** | 68.9% | 455.6 | 99.7% |

Table 6: Ablation study for 100 instances in 4 datasets with different guidances.

## A.10 HYPERPARAMETER FOR IP GUIDED SAMPLING

During the sampling process, we configured the number of steps to be 1000 for IP Guided Diffusion sampling (IP Guided DDPM) and 100 for Non-Markovian IP Guided Diffusion sampling (IP Guided DDIM). We provide the specific values for the gradient scale $s$ and leverage factor $\gamma$ in Table 7.

| dataset | IP Guided DDIM | | IP Guided DDPM | |
|---|---|---|---|---|
| | $s$ | $\gamma$ | $s$ | $\gamma$ |
| SC (small) | 100,000 | 0.9 | 15,000 | 0.1 |
| SC (medium) | 150,000 | 0.9 | 22,500 | 0.1 |
| SC (large) | 200,000 | 0.9 | 30,000 | 0.1 |
| CA | 20,000 | 0.7 | 10,000 | 0.3 |
| CF | 1,000 | 0.7 | 500,000 | 0.1 |
| IS | 20,000 | 0.5 | 10,000 | 0.1 |

Table 7: $s$ and $\gamma$ settings in different dataset

## A.11 HYPERPARAMETER TUNING EXPERIMENTS

In this experiment, we aim to investigate the effect of the gradient scale $s$ and leverage factor $\gamma$, as depicted in equation 7, on both IP guided sampling (IP Guided DDPM) and Non-Markovian IP guided sampling (IP Guided DDIM). We utilize the SC and CA datasets to calculate the average feasibility ratio (fea.) and average objective value (obj.) for each parameter group. The specific results for SC can be seen in Table 8 and 9, while the results for CA are presented in Table 10 and 11. In general, a larger $\gamma$ tends to lead to a better objective value but a lower feasibility ratio. However, this relationship does not hold true in all cases. As for the gradient scale $s$, its choices depend on the dataset and sampling algorithms employed.

| $s$ | $\gamma$ | obj. | fea. |
|---|---|---|---|
| 60,000 | 0.1 | 5611.9 | 100% |
| 60,000 | 0.3 | 2534.4 | 100% |
| 60,000 | 0.5 | 1097.2 | 100% |
| 60,000 | 0.7 | 1328.9 | 100% |
| 60,000 | 0.9 | 693.5 | 100% |
| 80,000 | 0.1 | 3108.1 | 100% |
| 80,000 | 0.3 | 2216.6 | 100% |
| 80,000 | 0.5 | 936.9 | 100% |
| 80,000 | 0.7 | 1167.1 | 100% |
| 80,000 | 0.9 | 605.3 | 99.5% |
| 100,000 | 0.1 | 39421.2 | 100% |
| 100,000 | 0.3 | 1462.5 | 100% |
| 100,000 | 0.5 | 834.7 | 100% |
| 100,000 | 0.7 | 1005.6 | 100% |
| 100,000 | 0.9 | 539.1 | 100% |

Table 8: IP Guided DDIM on the SC dataset.

| $s$ | $\gamma$ | obj. | fea. |
|---|---|---|---|
| 15,000 | 0.1 | 594.7 | 96.5% |
| 15,000 | 0.3 | 14663.8 | 100% |
| 15,000 | 0.5 | 3285.6 | 100% |
| 15,000 | 0.7 | 1285.6 | 97.0% |
| 15,000 | 0.9 | 784.3 | 86% |
| 20,000 | 0.1 | 915.1 | 100% |
| 20,000 | 0.3 | 10245.4 | 100% |
| 20,000 | 0.5 | 3385.9 | 100% |
| 20,000 | 0.7 | 2007.8 | 100% |
| 20,000 | 0.9 | 1418.6 | 100% |
| 25,000 | 0.1 | 29553.3 | 100% |
| 25,000 | 0.3 | 8560.6 | 100% |
| 25,000 | 0.5 | 4162.8 | 100% |
| 25,000 | 0.7 | 3288.83 | 100% |
| 25,000 | 0.9 | 2415.2 | 100% |

Table 9: IP Guided DDPM on the SC dataset.

## A.12 OTHER RELATED WORK

Numerous studies have explored the application of deep learning methods in solving Integer Programming (IP) problems, but these studies have different underlying approaches. In particular, Bengio et al. (2021) categorize existing methods into three main groups:

- Group 1: End-to-end learning involves training a machine learning model to directly generate solutions based on input instances. In the context of solving IP problems, this entails learning to construct solutions or improve existing solutions. Examples of construction methods in this group include Nair et al. (2020); Yoon (2022); Han et al. (2023), which have been discussed in Section 6. Another line in end-to-end learning focuses on learning to improve solutions, i.e., neighborhood search techniques (Hottung & Tierney, 2020; Song et al., 2020; Wu et al., 2021; Sonnerat et al., 2021). These methods learn to search

| $s$ | $\gamma$ | obj. | fea. |
|--------|-----|---------|-------|
| 10,000 | 0.1 | 5504.4  | 93.5% |
| 10,000 | 0.3 | 18856.2 | 97.5% |
| 10,000 | 0.5 | 23651.5 | 99.5% |
| 10,000 | 0.7 | 26039.0 | 98.0% |
| 10,000 | 0.9 | 28255.3 | 35.0% |
| 20,000 | 0.1 | 8280.9  | 97.5% |
| 20,000 | 0.3 | 21489.3 | 95.0% |
| 20,000 | 0.5 | 24618.1 | 95.5% |
| 20,000 | 0.7 | 26759.6 | 95.5% |
| 20,000 | 0.9 | 28105.7 | 24.0% |
| 30,000 | 0.1 | 10579.2 | 97.0% |
| 30,000 | 0.3 | 22771.0 | 93.0% |
| 30,000 | 0.5 | 26011.3 | 93.5% |
| 30,000 | 0.7 | 26622.5 | 90.5% |
| 30,000 | 0.9 | 28490.1 | 27.0% |

| $s$ | $\gamma$ | obj. | fea. |
|--------|-----|--------|-------|
| 5,000  | 0.1 | 1310.1 | 73.5% |
| 5,000  | 0.3 | 1518.5 | 58.0% |
| 5,000  | 0.5 | 1844.5 | 22.0% |
| 5,000  | 0.7 | 532.5  | 3.5%  |
| 5,000  | 0.9 | -      | 0.0%  |
| 10,000 | 0.1 | 575.2  | 92.5% |
| 10,000 | 0.3 | 782.5  | 86.5% |
| 10,000 | 0.5 | 1270.3 | 66.0% |
| 10,000 | 0.7 | 2126.5 | 17.5% |
| 10,000 | 0.9 | 923.3  | 0.5%  |
| 20,000 | 0.1 | 285.1  | 99.5% |
| 20,000 | 0.3 | 407.1  | 98.0% |
| 20,000 | 0.5 | 744.1  | 91.5% |
| 20,000 | 0.7 | 1430.5 | 57.5% |
| 20,000 | 0.9 | 1885.9 | 2.0%  |

Table 10:  IP Guided DDIM on the CA dataset.    Table 11: IP Guided DDPM on the CA dataset.

for high-quality feasible solutions in the neighborhood of given initial solutions in order to improve performance. It should be noted that these methods usually require a solver to obtain an initial solution.

- Group 2: Learning to configure algorithms involves using machine learning to select the values of hyperparameters in complex optimization algorithms. This can improve the efficiency of solving problems (Hutter et al., 2010).

- Group 3: Learning alongside optimization focuses on developing existing CO algorithms, particularly the branch-and-bound framework, that continuously utilize a machine learning model throughout their execution. For example, learning to branch (He et al., 2014; Khalil et al., 2016; Ding et al., 2020a; Balcan et al., 2018; Gupta et al., 2020) involves developing machine learning algorithms to generate policies for variable selection when expanding the branch and bound tree. Another works focus on learning to node selection (Khalil et al., 2022; He et al., 2014), where the machine learning model learns a score policy for "open" nodes in the branch-and-bound tree and selects suitable nodes to obtain better performance. Additionally, some approaches focus on generating cut planes to reduce the search space during the solving process, such as those proposed by Ding et al. (2020b) and Tang et al. (2020).

Our approach belongs to Group 1, specifically learning to construct solutions. The solutions generated by our method can be used as input for algorithms from other groups to further improve performance. Importantly, to the best of our knowledge, our approach is the first to generate complete and feasible solutions using pure neural techniques, without relying on any solvers. We refer readers to Bengio et al. (2021) for a more thorough review.

