# OpenReview forum: "Effective Generation of Feasible Solutions for Integer Programming via Guided Diffusion"
_ICLR.cc/2024/Conference — Submitted to ICLR 2024_

### Official Review · Reviewer_MZmK · 2023-10-28

**Soundness:** 3 good
**Presentation:** 2 fair
**Contribution:** 3 good
**Rating:** 6
**Confidence:** 4

**Summary:**

This paper studies the problem of finding feasible solutions to integer programming problems. The authors propose a novel framework that generates complete feasible solutions end-to-end. Their framework learns the embeddings for IP instances and their solutions and then uses diffusion models to learn the distributions. Finally, they perform sampling with trained models.

Key results: From their experimental results, it appears that their sampling methods provide solutions with a higher proportion of which are feasible solutions and have smaller objectives than our approaches.
Trained on small-size datasets, their models are able to scale to large-scale instances.

**Strengths:**

1. From their experimental results, it appears that their sampling methods provide solutions with a higher proportion of which are feasible solutions and have smaller objectives than our approaches.
2. Trained on small-size datasets, their models are able to scale to large-scale instances.

**Weaknesses:**

Major comments:

1.	“For SCIP, we adopt the first solution obtained through non-trivial heuristic algorithms during the solving phase.” I don’t know whether this comparison is fair. Did you try, for example, using the solutions they get within a fixed window of time?

2.	Why do you compare your algorithm mostly with SCIP instead of Gurobi which is possibly a much better solver.

3.	How does the objective value that you sampled compare to the optimal solution? How close are they? If they are far from each other, having a high feasible ratio does not mean anything. The feasible region increases exponentially, so there could be a large number of feasible solutions that are far from the optimal solution.

Minor comments:

1.	In “Related work”, you mentioned “our method aims to learn the latent structure …, without any reliance on the IP solver.”, but you still need to complete partial solutions use Completesol heuristic from SCIP.

2.	In page 8, the first paragraph, you mentioned “the coverage is set to 0.1 and 0.2 due to the difficulty in finding feasible partial solutions when C > 0.2.”. What do you mean by difficulty? Does it mean that you cannot find any feasible partial solutions within 30 generated solutions?

3.	Is it possible to generate repeated solutions so that the performance is not improving?
Possible typoes: Page 3 last paragraph: DDIM then

**Questions:**

Combined in the "Weaknesses"

---

> ### Author Response · Authors · 2023-11-20
> **Response to Reviewer MZmK**
>
> Thank you for reviewing our work! We try to address your major concerns as follows
>
> **Fair comparison and Gurobi:** The main goal of this paper is to generate good and complete initial feasible solutions end-to-end using neural network approaches, which is different from both SCIP and Gurobi solves: they aim for producing optimal solutions using initial feasible solutions as a starting point by using the branch and bound algorithm within a fixed time window. Hence, we compare the heuristic solutions obtained from SCIP and solutions after completing from Neural Diving. To further ensure the comparison comprehensive, in the paper, we have added additional results of the best heuristic solutions from Gurobi, as well as the best heuristic solutions from the PS+Gurobi algorithm, as suggested by Reviewer wgcj. The results show that the complete solutions generated by our methods have comparable quality to the best heuristic solutions from Gurobi in the SC dataset and better objectives in the CF and IS datasets. The partial solutions produced from our methods, combined with the CompleteSol heuristic, further improve the quality of solutions beyond all baseline methods. For more details, please refer to table 1 and table 2 in Section 5 in the updated manuscript (or check Table 1 and Table 2 in our global responses).
>
> **Small $C$ and Qualitative analysis:**
> We have to point out some misunderstanding in the reviewer's comments. Our algorithms do not require small $C$ to ensure feasibility because they have a high probability of generating complete solutions. Please refer to the result of IP Guided DDIM in table 1 in section 5, which shows that the complete solutions generated by our method alone have a feasibility ratio of at least 90%. Besides, in section 5.1, we provide an illustrative example to demonstrate the distinction between our methods and Neural Diving. The example highlights that IP Guided DDIM is capable of obtaining the optimal solution during sampling without the reliance of Solvers, whereas Neural Diving requires the Solver to complete the partial solution. In addition, we have discovered that randomly sampling a certain portion of variables from the generated solutions, with completion from the CompleteSol heuristic, further improves solution quality.
>
> Moreover, we have included qualitative analysis of the generatived solutions: we sampled 1000 solutions from a single instance of the SC and IS datasets and plotted the distribution of corresponding objectives, see Section 5.3 in the updated manuscript (or see [here](https://anonymous.4open.science/r/Guided_diffusion_for_IP-B4C0/SC_instance.pdf) for the distribution of SC instance and [here](https://anonymous.4open.science/r/Guided_diffusion_for_IP-B4C0/IS_instance.pdf) for the IS instance ). The majority of solutions (over 95%) from our methods have better quality than the Gurobi heuristic. Furthermore, when combining our methods with the CompleteSol heuristic,the distributions of solutions are even closer to the optimal values.
>
> We try to address your minor concerns as follows:
>
> * [Partial solution] No, our approach does not need CompleteSol to generate complete feasible solutions.
>
> * [Coverage ratio] We empirically observed that for the CF dataset, when the value of $C$ is set to above 0.2, the feasibility ratio of partial solutions obtained from Neural Diving is consistently low. On average, less than 40% of the 3000 samples (30*100 instances) were found to be feasible. When $C$ is set to 0.3, none of the generated solutions were found to be feasible.
>
> * [Repeated solutions] See our qualitative analysis.

---

> > ### Comment · Reviewer_MZmK · 2023-11-20
> >
> > I am grateful to the authors for their careful rebuttal. My comments are addressed. In view of that, I changed my rating upward by one point (from 5 to 6).

---

> > > ### Author Response · Authors · 2023-11-21
> > >
> > > Thank you again for your time and efforts. We appreciate your valuable comments to our paper.

---

### Official Review · Reviewer_wgcj · 2023-10-30

**Soundness:** 3 good
**Presentation:** 3 good
**Contribution:** 3 good
**Rating:** 6
**Confidence:** 4

**Summary:**

This paper proposes a novel framework that generates complete feasible solutions end-to-end (i.e., assigning all variables using neural networks) for Integer Programming (IP) problems, in contrast to most prior works that generate partial solutions (i.e., only assigning a subset of variables using neural networks).
Specifically, it proposes a contrastive learning approach to capture the relationship between the IP instances and the solutions, a diffusion model to generate solution embeddings, and a guided sampling strategy to enhance the feasibility and quality of solutions.
Experiments on four datasets show that the proposed method outperforms previous state-of-the-art methods in terms of feasible ratio and objective value.

**Strengths:**

1.	This work is well motivated and the paper is easy to follow.
2.	While most previous methods can only generate partial solutions, this work represents a valuable attempt to an end-to-end framework to generate complete feasible solution.
3.	Experiments demonstrate the effectiveness of the proposed method.

	a)	Experiments on four datasets demonstrate the effectiveness of the proposed methods compared with Neural Diving and SCIP in terms of feasible ratio and objective value.

	b)	The scalability test demonstrates that the proposed method can generalize to large instances.

	c)	The ablation study demonstrates the effectiveness of the IP guidance.

	d)	The authors also conduct hyperparameter tuning experiments to investigate the effect of the gradient scale $s$ and the leverage factor $\gamma$.

**Weaknesses:**

1.	The authors may want to add [1] as a baseline.
2.	The prediction loss defined in Eq. (3) empirically performs better than that from general diffusion models. It would be better to provide some intuitive interpretation. Moreover, the authors may want to provide the inference algorithm of the modified diffusion model.
3.	As diffusion generative models may suffer from inefficiency in both training and inference, the authors may want to report the training and inference time.

[1] Qingyu Han, Linxin Yang, Qian Chen, Xiang Zhou, Dong Zhang, Akang Wang, Ruoyu Sun, and Xiaodong Luo. A gnn-guided predict-and-search framework for mixed-integer linear programming. In The Eleventh International Conference on Learning Representations, 2023.

**Questions:**

1.	Is this work the first one to generate complete solutions?
2.	See Weakness 2. The training loss defined in Eq. (3) is different from general diffusion models. Does it cause a different inference algorithm?

---

> ### Author Response · Authors · 2023-11-20
> **Response to Reviewer wgcj**
>
> Thank you for reviewing our work!
>
> **More experiments:** We added more experimental results comparing our method with Gurobi and the baseline suggested by the reviewer. Please check the results in the Section 5 of the manuscript (or check Table 1 and Table 2 in our global responses). Results show that our methods consistently produce higher quality solutions than these two baselines.
>
> We also report toal training time and total sampling time for 3000 solutions on a workstation equipped with two Intel(R) Xeon(R) Platinum 8163 CPUs @ 2.50GHz, 176GB RAM, and two Nvidia V100 GPUs. During the inference phase, IP Guided DDIM exhibits faster performance than IP Guided DDPM, with average time of 0.46s-1.68s for sampling each solution (see Appendix A.8 or Table 3 in global responses for more detail).
>
>
> **Q1**:
> Yes, to the best of our knowledge, our approach is the first that generates complete and feasible solutions using pure neural techniques, without relying on any solvers.
>
> **Q2: loss and inference algorithm**
>
> The loss in Eq.(3) utilizes noisy embeddings to reconstruct the original embeddings as the objective. Intuitively, this helps enhance denoising capability in the neural networks, and facilitates the simultaneous training of the solution decoder via estimating $\mathbf{z} _{\mathbf{x}}$. Regarding the inference algorithm, as mentioned in Section 2, in the reverse process, the mean of $\mathbf{z} _{\mathbf{x}}^{(t-1)}$ can be approximated by adding $\mathbf{z} _{\mathbf{x}}^{(0)}$ as a condition
> $$
>   \mathbf{\mu} _{\theta}(\mathbf{z} _{\mathbf{x}}^{(t)},t) = \frac{\sqrt{\alpha _t}(1-\bar{\alpha} _{t-1})}{1-\bar{\alpha} _t}\mathbf{z} _{\mathbf{x}}^{(t)}+\frac{\sqrt{\bar{\alpha} _{t-1}}\beta _t}{1-\bar{\alpha} _t}\mathbf{z} _{\mathbf{x}}^{(0)} \tag{1}
> $$
> In paper [1], they use the following formula (2) from the forward process
> $$
>     \mathbf{z} _{\mathbf{x}}^{(t)} = \sqrt{\bar{\alpha} _t} \mathbf{z} _{\mathbf{x}}^{(0)}+\sqrt{1-\bar{\alpha} _t} \mathbf{\epsilon} \tag{2}
> $$
> to replace $\mathbf{z} _{\mathbf{x}}^{(0)}$.  It implies that
> $$
>             \mathbf{\mu} _{\theta}(\mathbf{z} _{\mathbf{x}}^{(t)},t) = \frac{\sqrt{\alpha _t}(1-\bar{\alpha} _{t-1})}{1-\bar{\alpha} _t}\mathbf{z} _{\mathbf{x}}^{(t)}+\frac{\sqrt{\bar{\alpha} _{t-1}}\beta_t}{1-\bar{\alpha} _t}\mathbf{z} _{\mathbf{x}}^{(0)}
>             = \frac{\sqrt{\alpha _t}(1-\bar{\alpha} _{t-1})}{1-\bar{\alpha} _t}\mathbf{z} _{\mathbf{x}}^{(t)}+\frac{\sqrt{\bar{\alpha} _{t-1}}\beta _t}{(1-\bar{\alpha} _t)\sqrt{\bar{\alpha} _t}}(\mathbf{z} _{\mathbf{x}}^{(t)}- \sqrt{1-\bar{\alpha} _t} \mathbf{\epsilon})
>             = \frac{1}{\sqrt{\alpha _t}} (\mathbf{z} _{\mathbf{x}}^{(t)}- \frac{1-\alpha _t}{\sqrt{1-\bar{\alpha} _t}} \mathbf{\epsilon)}.
> $$
> The last equation holds because $\alpha _t:=1-\beta _t$ and $\bar{\alpha} := \prod _{s=1}^t \alpha _s$.  Therefore, we obtain the original sampling method of DDPM (Ho et al., 2020).  In our work, since we use the neural network to predict $\mathbf{z} _{\mathbf{x}}^{(0)}$,  we can directly use Eq.(1) to estimate the mean of $\mathbf{z} _{\mathbf{x}}^{(t-1)}$ in sampling phase as shown in Appendix A.4 in the updated manuscript. In the experiments, we use the same variance estimation as DDPM (see Eq.(7) in Ho et al. (2020)). Based on Eq.(2), we also produce the estimation of $\mathbf{\epsilon}$ by using $\mathbf{z} _{\mathbf{x}}^{(t)}$ and predicted $\mathbf{z} _{\mathbf{x}}^{(0)}$ and use it in the inference phase for DDIM.
>
> [1]. Jonathan Ho, Ajay Jain, and Pieter Abbeel. Denoising diffusion probabilistic models. Advances in neural information processing systems, 33:6840-6851, 2020.

---

> > ### Comment · Reviewer_wgcj · 2023-11-22
> >
> > Thanks for the authors' responses, which have addressed most of my concerns. However, I think the observed performance improvement, while notable, is not yet substantial enough for a higher score. Overall, I would like to keep my score.

---

> > > ### Author Response · Authors · 2023-11-23
> > >
> > > Thanks again! We are grateful to your valuable suggestions to our paper.

---

### Official Review · Reviewer_jBX3 · 2023-10-31

**Soundness:** 2 fair
**Presentation:** 2 fair
**Contribution:** 2 fair
**Rating:** 5
**Confidence:** 3

**Summary:**

A solution generation method is adopted to estimate binary solutions of integer programming. The method includes a contrastive learning  gaining initial representations of solutions and instances, and a conditioned generative model estimating binary solutions. Guided sampling is adapted from present diffusion models to increase the feasibility ratio.

**Strengths:**

Applying cutting-edge deep learning to solve integer programming problems is encouraging. This research focuses on generating feasible solutions by generative model and borrows the powerful representation learning capability of neural networks. The method is technically sound by simply applying contrastive learning and diffusion model for solution estimation.

**Weaknesses:**

My first concern is the insufficient comparison in experiments. As described in related work, considerable literature attempted to improve the diving method in solvers. Except Neural Diving (Nair et al., 2020), many follow-up works continue similar research topics. More recent methods should be compared. Even by only comparing Neural Diving, the results are not enough. The training time and resource usage are not clear, which is important to show practicality and efficiency of applying multiple deep neural networks in the proposed method. Moreover, the functions of contrastive model and generative model are not showcased by ablation study.

Many works apply deep learning methods to solve integer programming problems with totally feasible solutions. To name a few, "A general large neighborhood search framework for solving integer linear programs", "Learning large neighborhood search policy for integer programming", "Mip-gnn: A data-driven framework for guiding combinatorial solvers". The advantage of this research over this line of works is not clear. The use case of the given method is not given. Many descriptions are not well explained (see questions).

-----------------After rebuttal-------------------------

I appreciate authors' detailed rebuttal. I still think the novelty is not high. The results and literature added in rebuttal are important and should have been in the original version. I raised my score a bit to 5.

**Questions:**

1. Why SCIP is chosen in experiments but not Gurobi, given the fact that Gurobi often performs better than SCIP.
2. GCN is described by "It does not explicitly incorporate objective and constraint information during sampling, often resulting in infeasible complete solutions." In Gasse et al. (2019), GCN always gains feasible solutions.
3. Any integer programming problem can be converted into a 0-1 programming. But the conversion increases the number of constraints a lot. How large integer programming can the method solve?
4. What is the advantage of contrastive learning compared to supervised learning? Additional experiment should be provided to see the effect of contrastive learning without labeled solutions

---

> ### Author Response · Authors · 2023-11-20
> **Response to Reviewer jBX3 part 1**
>
> Thank you for reviewing our paper! We try to address your concerns and answer the raised questions below.
>
> **First concern and Q1: More empirical comparisons.** We added more experimental results comparing our method with Gurobi and a more recent baseline method (Han et al., 2023) suggested by Reviewer wgcj, which shows that our methods consistently produce higher quality solutions than these two baselines. Please check Table 1 and Table 2 in Section 5 for more detail (or check Table 1 and Table 2 in our global response).
>
> **Other concerns and Q4: ablations and training time.** Further, we ablated on the contrastive learning, i.e., with and without contrastive learning for the embeddings. Specifically, we include an ablation experiment in which we train IP and solution embeddings directly via the training produce of diffusion and decoder (Algorithm 2 in appendix A.3 in the updated manuscript) without using CISP, the results were updated in the Appendix A.9 of the updated manuscript (or check Table 4 in the global responses ).
>
> The results show that CISP plays a crucial role in ensuring that the solutions produced by our methods are more feasible. We found that the advantage of contrastive learning is its ability to extract meaningful representations for IP instances and solutions, which is achieved by the assumption that instances should stay close to their feasible solutions and away from their infeasible ones. This can further integrate features from different forms, as the instance is represented using a bipartite graph and the solution is represented in a vector space. In contrast, this cannot be simply achieved by supervised learning.
>
> As for the training time and computation usage, all our evaluations were conducted on a workstation equipped with two Intel(R) Xeon(R) Platinum 8163 CPUs @ 2.50GHz, 176GB RAM, and two Nvidia V100 GPUs. We have provided the total training time and the total inference time for generating 3000 solutions below (also see appendix A.8 in the updated manuscript). During the inference phase, IP Guided DDIM exhibits faster performance than IP Guided DDPM, with average time of 0.46s-1.68s for sampling each solution (see Appendix A.8 or Table 3 in global responses for more detail).
>
>
> **Q2.**
> We think there is misunderstanding of some parts of our work possibly due to some misleading discussions in our paper. We agree with the reviewer that Gasse et al. (2019) proposed using a bipartite graph structure to model an IP instance and applied GCN to extract variable representations. But their primary focus was on learning the branching policy, i.e., selecting the variable for partitioning the node's search space. Follow-up studies, such as Nair et al. (2020), Yoon Taehyun (2022) and Han et al. (2023) confirm that directly using the bipartite graph and GCN to learn solutions for an IP instance may result in infeasible solutions, and thus extended this method by using partial assignments and CompleteSol heuristic to obtain complete feasible solutions. We clarified this in the revised manuscript in the paragraph 2 of Section 1.
>
> **Q3.**
> We agree with the review that the conversion increases the number of constraints. But the point we tried to make is that our approach is generic enough to solve more general integer programming problems. In fact, with the aforementioned computation space, our approach can solve sizable IP problems: the largest instance (CF dataset) solved by our approach has about 5000 variables and 5000 constraints. These sizes can be further improved with more computation resources and more pre-training on typical small-size IP instances.
>
> **References for part 1:**
>
> [1] Qingyu Han, Linxin Yang, Qian Chen, Xiang Zhou, Dong Zhang, Akang Wang, Ruoyu Sun, and Xiaodong Luo. A GNN-guided predict-and-search framework for mixed-integer linear programming. In International Conference on Learning Representations, 2023.
>
> [2] Maxime Gasse, Didier Chetelat, Nicola Ferroni, Laurent Charlin, and Andrea Lodi. Exact combinatorial optimization with graph convolutional neural networks. Advances in Neural Information Processing Systems, 32, 2019.
>
> [3] Vinod Nair, Sergey Bartunov, Felix Gimeno, Ingrid Von Glehn, Pawel Lichocki, Ivan Lobov, Brendan O’Donoghue, Nicolas Sonnerat, Christian Tjandraatmadja, Pengming Wang, et al. Solving mixed integer programs using neural networks. arXiv preprint arXiv:2012.13349, 2020.
>
> [4] Taehyun Yoon. Confidence threshold neural diving. CoRR, abs/2202.07506, 2022. URL https://arxiv.org/abs/2202.07506.

---

> ### Author Response · Authors · 2023-11-20
> **Response to Reviewer jBX3 part 2**
>
> **Other related works.**
> We thank the reviewer for sharing many papers. However, it appears that there may have been some misunderstanding regarding the relationship between our work and other papers. Therefore, we would like to provide further context in order to clarify our position and help the reviewer better understand our contribution.
>
> It is true that numerous works have been undertaken to apply deep learning methods to solve Integer Programming problems. But the underlying ideas are quite different to each other. Particularly, Bengio et al. (2021) categorize existing methods into three main groups (see Appendix A.12 in our update manuscript for a detailed discussion):
>
> * Group 1 End-to-end learning, which refers to the training of a machine learning model to directly generate solutions based on input instances. In the context of solving integer programming (IP) problems, this involves learning to construct solutions, as demonstrated by methods like Neural Diving (Nair et al.,2020; Taehyun Yoon, 2022) and Predict-Search framework (Han et al., 2023). However, **these methods still need to rely on a solver to generate complete solution**. Another line in end-to-end learning focuses on learning to improve solutions, i.e., neighborhood search techniques (Sonnerat et al., 2021; Wu et al., 2021). **It is noteworthy that these methods typically require a solver to acquire an initial solution**.
>
> * Group 2 Complex optimization algorithms usually have a set of hyper-parameters left constant during optimization. This area use machine learning to select the values of hyper-parameters.
>
> * Group 3 Learning alongside optimization:  This field focuses on developing existed CO algorithms, typically the branch-and-bound framework, that continuously utilize a machine learning model throughout their execution, including techniques such as learning to branch (Gasse et al., 2019) and learning to node selection (Khalil et al., 2022). These works aim to generate high-quality solutions by combining ML method with the branch-and-bound framework in solvers.
>
> Our approach falls into learning to construct solutions of Group 1. Importantly, to the best of our knowledge, our approach is the first that generates complete and feasible solutions using pure neural techniques, without relying on any solvers.
>
> The papers mentioned by the review fall into other groups and study different research questions. Specifically, the papers (Sonnerat et al., 2021; Wu et al., 2021) falls into learning to improve solutions of Group 1 since they focus on learning methods for enhancing solution quality using neighborhood search techniques, not directly related to solution generation. The papers Khalil et al., (2022) falls more into Group 3 and primarily focuses on node selection and also proposes the policy of producing a partial solution through a prescribed rounding threshold, which is then completed using SCIP, similar to Nair et al. (2020) and Taehyun Yoon (2022).
>
>
>
> **References for part 2:**
>
> [1] Qingyu Han, Linxin Yang, Qian Chen, Xiang Zhou, Dong Zhang, Akang Wang, Ruoyu Sun, and Xiaodong Luo. A GNN-guided predict-and-search framework for mixed-integer linear programming. In International Conference on Learning Representations, 2023.
>
> [2] Yoshua Bengio, Andrea Lodi, and Antoine Prouvost. Machine learning for combinatorial optimization: a methodological tour d’horizon. European Journal of Operational Research, 290(2):405–421, 2021.
>
> [3] Vinod Nair, Sergey Bartunov, Felix Gimeno, Ingrid Von Glehn, Pawel Lichocki, Ivan Lobov, Brendan O’Donoghue, Nicolas Sonnerat, Christian Tjandraatmadja, Pengming Wang, et al. Solving mixed integer programs using neural networks. arXiv preprint arXiv:2012.13349, 2020.
>
> [4] Taehyun Yoon. Confidence threshold neural diving. CoRR, abs/2202.07506, 2022. URL https://arxiv.org/abs/2202.07506.
>
> [5] Nicolas Sonnerat, Pengming Wang, Ira Ktena, Sergey Bartunov, and Vinod Nair. Learning a large neighborhood search algorithm for mixed integer programs. arXiv preprint arXiv:2107.10201,2021.
>
> [6] Yaoxin Wu, Wen Song, Zhiguang Cao, and Jie Zhang. Learning large neighborhood search policy for integer programming. Advances in Neural Information Processing Systems, 34:30075–30087, 2021.
>
> [7] Maxime Gasse, Didier Chetelat, Nicola Ferroni, Laurent Charlin, and Andrea Lodi. Exact combinatorial optimization with graph convolutional neural networks. Advances in Neural Information Processing Systems, 32, 2019.
>
> [8] Elias B Khalil, Christopher Morris, and Andrea Lodi. Mip-gnn: A data driven framework for guiding combinatorial solvers. In Proceedings of the AAAI Conference on Artificial Intelligence, volume 36, pp. 10219–10227, 2022.

---

> ### Author Response · Authors · 2023-11-21
>
> Dear Reviewer jBX3, thank you again for your time and efforts. We appreciate your constructive suggestions to our paper. We hope our response can address your concerns and would like to hear your feedback again. Please forgive our eagerness and impatience, we are very keen to improve our paper and really appreciate the response from an expert like you!

---

### Official Review · Reviewer_CS3k · 2023-11-10

**Soundness:** 3 good
**Presentation:** 3 good
**Contribution:** 2 fair
**Rating:** 8
**Confidence:** 2

**Summary:**

The paper proposed a learning based IP solver. For problem and solution embedding, the solver took the GCN framework, combined with contrastive learning inspired by CLIP. In addition the authors adapted DDPM/DDIM by introducing IP specific guidance into the sampling procedure. Experiments on several IP problems showed superior performance to both Neural Diving and SCIP.

**Strengths:**

Several key components were designed to make the solver specifically effective for IP. Experiments are solid.

**Weaknesses:**

To better validate that the quality of the proposed solver, comparison between the found optimal objective value and the ground-truth (global optimum) would be more convincing, the paper only provided relative comparison between the proposed solver and two baseline approaches.

**Questions:**

N/A

---

> ### Author Response · Authors · 2023-11-20
> **Response to Reviewer CS3k**
>
> Thank you for your suggestion! We have employed Gurobi to solve each dataset instance for 100 seconds, aiming to obtain the best possible solutions as optimal values. In the updated manuscript, we have included the average optimal values for each dataset in Table 1 and Table 2 in Section 5 (we also reported these results in Table 1 and Table 2 in the global responses). The experimental results demonstrate that our feasible solutions achieve a gap of 7% to 34% compared to the optimal values. This performance is superior to the heuristic methods from Gurobi and SCIP.

---

### Author Response · Authors · 2023-11-20
**Global responses to all reviewers**

We sincerely thank all the reviewers for their feedback and constructive comments. We have made several changes to the paper based on their suggestions.

* **More baselines.** We conducted additional experiments to compare our method with the best heuristic results from the commercial solver Gurobi and the recent baseline method PS+Gurobi (Han et al., 2023) as suggested by Reviewers jBX3, wgcj, and MZmK. We used Gurobi to solve each dataset instance for 100 seconds to obtain the best possible solutions as optimal values for better validating the effectiveness of our method as suggested by Reviewer CS3k. The results of these experiments are now included in table 1 and table 2 of Section 5 in the updated manuscript. For convenience, we also report the specific values here:

| Instance | IP Guided DDIM | IP Guided DDPM | IP Guided DDIM + Completesol | ND (low coverage) + Completesol | PS + Gurobi | SCIP | Gurobi | Optimal |
| --- | --- | --- | --- | --- | --- | --- | --- | --- |
| SC (min) | 533.5 (99.8%) | 577.9 (95.7%) | **255.5 (100%)** | 849.0 (100%) | 593.7| 1967.0 | 522.4 | 168.28 |
| CA (max) | 26916.9 (97.1%) | 800.3 (87.3%) | **32491.1 (99.7%)** | 30143.6 (87.0%) | 31159.5 | 28007.4 | 30052.0| 36102.6 |
| CF (min) | 25119.2 (89.7%) | 58488.1 (44.0%) | **14224.1 (100%)** | 14259.8 (81.3%) | 32119.8 | 84748.4 | 50397.3 | 11405.5 |
| IS (max) | 455.6 (99.7%) | 129.9 (**100%**) | **639.4 (100%)** | 484.1 (90.4%) | 587.9| 447.8 | 415.5 | 685.3 |

Table 1: The average objective value (feasible ratio) for 100 instances.

| Size | IP Guided DDIM | IP Guided DDPM | IP Guided DDIM + Completesol | ND (low coverage) + Completesol | PS + Gurobi | SCIP | Gurobi | Optimal |
| --- | --- | --- | --- | --- | --- | --- | --- | --- |
| small | 533.5 (99.8%) | 594.7 (96.5%) | **255.5 (100%)** | 849.0 (**100%**) | 593.7 | 1967.0 | 522.4 | 168.3 |
| medium | 486.8 (99.9%) | 451.8 (83.7%) | **217.4 (100%)** | 1145.8 (**100%**) | 737.0 | 2236.2 | 718.8 | 140.4 |
| large | 464.9 (**100%**) | 440.7 (77.9%) | **195.9 (100%)** | 1465.6 (**100%**) | 994.9 | 2386.0 | 1454.5 | 126.9 |

Table 2: The average objective value (feasible ratio) for 100 instances in 3 different size SC datasets.

* **Training and inference time.** Our all evaluations are preformed in a machine with two Intel(R) Xeon(R) Platinum 8163 CPU @ 2.50GHz, 176GB ram and two Nvidia V100 GPUs. We reported the total training and inference time for sampling 3000 solutions in Appendix A.8 as suggested by Reviewers jBX3 and wgcj. The specific results are

| Dataset | Training (CISP + Diffusion) | IP Guided DDIM | IP Guided DDPM |
| ------- | -------------- | -------------- | -------------- |
| SC      | 24.4m | 37.5m| 374m  |
| CA     | 9.3m  | 23m  | 233.5m        |
| CF     | 71.7m  | 84m  | 805m          |
| IS      | 11.1m  | 23m | 234m          |

Table 3: Total training time and total time for sampling 3000 solutions for each dataset.

* **Ablation study on CISP.** For presenting the advantage of contrastive learning empirically, we add an experiment in which we train IP and solution embeddings directly via the training produce of diffusion and decoder without using CISP the results were presented in Appendix A.9 in the updated manuscript. The specific values are

| dataset | Unguided DDIM | Constraint Guided DDIM | Objective Guided DDIM | IP Guided DDIM w/o CISP | IP Guided DDIM |
| --- | --- | --- | --- | --- | --- |
| SC (min) | - (0%) | 63046.9 (99.8%) | - (0%) | 763.4 (99.8%) | **533.5 (99.8%)** |
| CA (max) | - (0%) | 5157.2 (**99.7%**) | - (0%) | 23383.3 (57.7%) | **26916.9** (97.1%) |
| CF (min) | - (0%) | 53311.2 (74.1%) | - (0%) | 31319.8 (41.7%) | **25119.2 (89.7%)** |
| IS (max) | - (0%) | 386.5 (**100%**) | - (0%) | **479.1** (68.9%) | 455.6 (99.7%) |

Table 4: The average objective value (feasible ratio) for ablation study on 100 instances in 4 datasets with different guidances.


* **The qualitative analysis of solutions.** We also included qualitative analysis of the generative solutions: we sampled 1000 solutions from a single instance of the SC and IS datasets and plotted the distribution of corresponding objectives, see Section 5.3 in the updated manuscript. The distribution of SC instance can be accessed [here](https://anonymous.4open.science/r/Guided_diffusion_for_IP-B4C0/SC_instance.pdf) and the IS instance distribution can be found [here](https://anonymous.4open.science/r/Guided_diffusion_for_IP-B4C0/IS_instance.pdf).

* **Others.** We have added section on related works in the field of machine learning for solving Integer Programming for readers to better understand our work in Appendix A.12. Additionally, we have corrected typos and improved the overall readability in the newly uploaded version.

[1] Qingyu Han, Linxin Yang, Qian Chen, Xiang Zhou, Dong Zhang, Akang Wang, Ruoyu Sun, and Xiaodong Luo. A GNN-guided predict-and-search framework for mixed-integer linear programming. In International Conference on Learning Representations, 2023.

---

### Meta-Review · Area_Chair_Jrjj · 2023-12-05

**Metareview:**

The authors propose a diffusion-based solution for the important problem of integer programming. From a novelty perspective, the idea is incremental, as they deploy standard denoising diffusion mechanisms. However, the diffusion is conducted in a latent space represented by an IP encoder. The generative method samples from the latent space and projects to the solution space using a solution decoder. Overall, while the novelty is limited from a diffusion angle, the paper is an interesting contribution to the Integer Programming aspect. In addition, the results are competitive against prominent IP solver methods. However, it seems that the Completesol heuristic is a decisive factor in the empirical results, to help refine the partial solutions generated by the diffusion process. It is a negative outcome of the paper that the diffusion mechanism cannot solve the IP problem without relying on a completion heuristic. The authors do not analyze in greater depth the limitation of the diffusion mechanism, and what should be done to ameliorate the weakness. Therefore, I recommend rejection in the current state.

**Justification For Why Not Higher Score:**

The limiting factor is that the proposed diffusion mechanism is not competitive enough without a completion solution heuristic.

**Justification For Why Not Lower Score:**

N/A

---

### Decision · Program_Chairs · 2024-01-16

Reject